# Vitamin D constrains inflammation by modulating the expression of key genes on Chr17q12-21.1

Ayse Kilic[1†], Arda Halu[1*†], Margherita De Marzio[1,2], Enrico Maiorino[1], Melody G Duvall[3], Thayse Regina Bruggemann[3], Joselyn J Rojas Quintero[3], Robert Chase[1], Hooman Mirzakhani[1], Ayse Özge Sungur[4,5], Janine Koepke[5], Taiji Nakano[6], Hong Yong Peh[3], Nandini Krishnamoorthy[3], Raja-Elie Abdulnour[3], Katia Georgopoulos[7], Augusto A Litonjua[8], Marie Demay[9], Harald Renz[10,11], Bruce D Levy[3], Scott T Weiss[1*†]

[1]Channing Division of Network Medicine, Department of Medicine, Brigham and Women's Hospital and Harvard Medical School, Boston, United States; [2]Department of Environmental Health, Harvard TH Chan School of Public Health, Boston, United States; [3]Division of Pulmonary and Critical Care Medicine, Department of Medicine, Brigham and Women's Hospital and Harvard Medical School, Boston, United States; [4]Behavioral Neuroscience, Experimental and Biological Psychology, Philipps-University, Marburg, Germany; [5]Excellence Cluster Cardio-Pulmonary System (ECCPS), Justus Liebig University Giessen, Giessen, Germany; [6]Department of Pediatrics, Graduate School of Medicine, Chiba University, Chiba, Japan; [7]Cutaneous Biology Research Center, Massachusetts General Hospital and Harvard Medical School, Boston, United States; [8]Division of Pediatric Pulmonary Medicine, Golisano Children's Hospital at Strong, University of Rochester Medical Center, Rochester, United States; [9]Endocrine Unit, Massachusetts General Hospital and Harvard Medical School, Boston, United States; [10]Institute of Laboratory Medicine and Pathobiochemistry, Molecular Diagnostics, Philipps University of Marburg and German Center for Lung Research (DZL), Marburg, Germany; [11]Department of Clinical Immunology and Allergology, Laboratory of Immunopathology Sechenov University, Moscow, Russian Federation

*For correspondence:
arda.halu@channing.harvard.edu (AH);
restw@channing.harvard.edu (STW)

†These authors contributed equally to the manuscript.

**Abstract** Vitamin D possesses immunomodulatory functions and vitamin D deficiency has been associated with the rise in chronic inflammatory diseases, including asthma (Litonjua and Weiss, 2007). Vitamin D supplementation studies do not provide insight into the molecular genetic mechanisms of vitamin D-mediated immunoregulation. Here, we provide evidence for vitamin D regulation of two human chromosomal loci, Chr17q12-21.1 and Chr17q21.2, reliably associated with autoimmune and chronic inflammatory diseases. We demonstrate increased vitamin D receptor (*Vdr*) expression in mouse lung CD4+ Th2 cells, differential expression of Chr17q12-21.1 and Chr17q21.2 genes in Th2 cells based on vitamin D status and identify the IL-2/Stat5 pathway as a target of vitamin D signaling. Vitamin D deficiency caused severe lung inflammation after allergen challenge in mice that was prevented by long-term prenatal vitamin D supplementation. Mechanistically, vitamin D induced the expression of the *Ikzf3*-encoded protein Aiolos to suppress IL-2 signaling and ameliorate cytokine production in Th2 cells. These translational findings demonstrate mechanisms for the immune protective effect of vitamin D in allergic lung inflammation with a strong molecular genetic link to the regulation of both Chr17q12-21.1 and Chr17q21.2 genes and suggest further functional studies and interventional strategies for long-term prevention of asthma and other autoimmune disorders.

## eLife assessment

The effect of Vitamin D supplementation in reducing asthma via anti-inflammatory mechanisms is a topic of wide interest, with somewhat conflicting published data. Here, bioinformatic approaches help to identify a role of VDR in inducing the expression of the key regulator Ikzf3, which possibly suppresses the IL-2/STAT5 axis, consequently blunting the Th2 response and mitigating allergic airway inflammation. These are **important** findings based on **convincing** evidence.

## Introduction

The vitamin D receptor was genetically associated with asthma in 2004 (*Raby et al., 2004*) and high intake of vitamin D by pregnant women is associated with about a 50% reduction in asthma risk in the mother's offspring (*Camargo et al., 2007*; *Devereux et al., 2007*). Together, these findings led to a comprehensive theory about how vitamin D deficiency could influence asthma occurrence through its effects on lung and immune system development and that progressive decreases in vitamin D intake from 1946 onward could be contributing to the epidemic of allergic and autoimmune diseases (*Litonjua and Weiss, 2007*). Vitamin D deficiency is the most common vitamin deficiency in the world today and is particularly prevalent in pregnant women where the fetal lung and immune system are developing in utero (*van der Pligt et al., 2018*; *Roth et al., 2018*; *Cashman, 2020*).

To determine the impact of vitamin D supplementation, we have previously performed a clinical trial, the Vitamin D Antenatal Asthma Trial (VDAART) in pregnant women who either had allergies or asthma or had family members with these conditions. The participants were given either 4400 IU of vitamin D3 or 400 IU and were followed throughout their pregnancy and for the first 6 years of the life of the child (*Litonjua et al., 2014*). The results of the trial were not statistically significant using conventional intent to treat analysis (*Litonjua et al., 2016*; *Litonjua et al., 2020*) but the reasons were complex, as is often seen when the effects of nutrients are studied. Most importantly, unlike conventional drug trials where one compares drug to no drug or alternative drug, in nutrient trials there is nutrient already present in the placebo group. This creates misclassification that can reduce trial power. When we performed a meta-analysis of the two pregnancy related trials of vitamin D, we got a statistically significant reduction in asthma in the offspring of women who had the higher vitamin D intake during pregnancy and, when we adjusted for the baseline level of vitamin D in the meta-analysis, we got a reduction in asthma risk of 50%, exactly what we saw in the observational studies performed previously (*Wolsk et al., 2017a*; *Wolsk et al., 2017b*).

To link the results of the VDAART trial directly to the chr17q12 locus, our group then genotyped the single-nucleotide polymorphism (SNP) rs12936231, located in ZPBP2, the gene adjacent to ORMDL3, and stratified the VDAART trial results by maternal genotype at this locus. This study found that the vitamin D effect in the trial was significantly influenced by genotype, with the GG genotype exhibiting a protective effect on asthma risk in the child and the CC or GC genotype not being responsive to vitamin D; additionally, this result was related to sphingolipid production with the children protected from asthma having higher sphingolipid production (*Kelly et al., 2019*).

This SNP (rs12936231) is one of two SNPs (the other being rs4065275) that alter the chromatin state of a regulatory domain, specifically two CTCF-binding sites, in the chr17q12-21.1 locus that are correlated with the expression of ORMDL3 (*Schmiedel et al., 2016*; *Verlaan et al., 2009*). 4C-seq assays previously demonstrated that the ORMDL3 promoter interacts with a long-range enhancer in IKZF3 that promotes (or represses) transcription of ORMDL3 in cells expressing both genes and the binding of CTCF per the G allele of rs12936231 in ZPBP2 blocks this interaction, resulting in reduced transcription of ORMDL3 on haplotypes with the rs12936231-G allele (*Schmiedel et al., 2016*). If rs12936231 is not activated, then there is activation of the other CTCF-binding site (rs4065275) intronic to ORMDL3, thus favoring the expression of ORMDL3 and increased asthma risk (*Schmiedel et al., 2016*).

Mouse models have contributed to our understanding of ORMDL3 function as mice expressing the ORMDL3 transgene exhibit spontaneous increases in airway hyper responsiveness (AHR), the essential feature of asthma. This increase in AHR was associated with airway remodeling and peri bronchial fibrosis without airway inflammation (*Miller et al., 2014*). ORMDL3 is pleotropic, influencing $Ca^{2+}$ signaling, and the unfolded protein response. Most importantly, increased expression of ORMDL3 inhibits the enzyme serine palmityl-transferase, the rate limiting step in the production

of sphingolipids, and ORMDL3 transgenic mice have reduced levels of sphingolipids (*Zhang et al., 2019*). What has been unclear so far is how ORMDL3 is controlled.

In the current manuscript, we provide evidence for vitamin D regulation of two human chromosomal loci, Chr17q12-21.1 and Chr17q21.2, reliably associated with autoimmune and chronic inflammatory diseases. We demonstrate increased vitamin D receptor (*Vdr*) expression in mouse lung CD4+ Th2 cells, differential expression of these Chr17q12-21.1 and Chr17q21.2 genes in Th2 cells based on vitamin D status and identify the IL-2/Stat5 pathway as a target of vitamin D signaling. Vitamin D induced the expression of the *Ikzf3*-encoded protein Aiolos to suppress IL-2 signaling and ameliorate cytokine production in Th2 cells thus demonstrating a mechanism for how vitamin D influences autoimmunity.

## Results

### Integrative bioinformatics identifies the potential control of key asthma and autoimmune loci in 17q12-21.1 and 17q21.2 by vitamin D

To address the genetic and molecular mechanisms by which vitamin D can influence asthma, allergic, and autoimmune disease risk, we studied the genetic loci of two chromosome regions of interest: (1) the *17q12-21.1* region, that includes the well-known asthma-associated *ORMDL3* gene (*Bouzigon et al., 2008*; *Moffatt et al., 2007*), and (2) the *17q21.2* region, that includes the *STAT5* gene, known to be extensively associated with autoimmune diseases (*Litherland et al., 2005*; *Kara et al., 2019*). We first examined the entire set of significant disease and trait associations in the *Chr17q12-21.1* (181 associations) and *Chr17q21.2* (100 associations) loci using the NHGRI-EBI GWAS Catalog (*Buniello et al., 2019*). Both regions had multiple Th1 and Th2 diseases associations with at least one significant SNP. *Chr17q12-21.1* had the highest number of associations related to Th2 diseases (*Figure 1A*) and *17q21.2* had the highest number of associations related to Th1 diseases (*Figure 1—figure supplement 1A*). To address the possible link to vitamin D intake, we next investigated a potential overlap of these SNPs with *VDR*-binding sites. Out of the 169 *VDR*-binding sites on Chr17, seven were in the *17q12-21.1* locus, concentrated in two bands on or near *IKZF3* and *ZPBP2* (*Figure 1B*). These *VDR*-binding sites also overlapped with *RXRA*-binding sites (*Figure 1B*). To assess their functional relevance, we searched for expression quantitative trait loci (eQTLs) in these *VDR*-binding regions. We found that four cis-eQTLs control *IKZF3* expression in whole blood and EBV-transformed lymphocytes: rs2941522 and rs12946510 in the enhancer region targeting *ORMDL3* and *IKZF3*, and rs1453559 and rs35564481 in the enhancer region targeting *ORMDL3*, *GSDMA*, and *GSDMB* (*Figure 1B*). Rs1453559 and rs2941522 have previous Genome-wide association studies (GWAS) associations with both asthma and autoimmune diseases (*Syreeni et al., 2021*; *Shrine et al., 2019*; *Marinho et al., 2012*), rs12946510 has previous GWAS associations with autoimmune diseases (*Hitomi et al., 2017*; *Keshari et al., 2016*), and rs35564481 has no known GWAS associations. For the *17q21.2* locus, the only *VDR*-binding site resides near the gene *PSMC3IP* in an enhancer site targeting *STAT5A* (*Figure 1—figure supplement 1B*). *PSMC3IP* expression is controlled by three eQTLs overlapping with this *VDR*-binding site (rs4793244, rs62078362, and rs111708606), none of which have any known disease associations. These results suggest that vitamin D binding is critical in this genomic region.

We next examined the linkage disequilibrium (LD) pattern between the Th1/Th2-associated SNPs and the SNPs in the *VDR*-binding sites (*Figure 1C*). Out of the seven SNPs with *VDR*-binding sites identified in *17q12-21.1* and *17q21.2*, the first four in *17q12-21.1* (rs2941522, rs12946510, rs35564481, and rs1453559) were in a high-LD block with many other Th1- to Th2-associated SNPs in the same region. These same four SNPs in *17q12-21.1* were also in high-LD with two SNPs in *ZPBP2* and *ORMDL3* (rs12936231 and rs4065275, respectively) that were previously shown to be functionally relevant to asthma (*Kelly et al., 2019*; *Schmiedel et al., 2016*; *Verlaan et al., 2009*). Additionally, two of the three SNPs with *VDR*-binding sites in *17q21.2* (rs4793244 and rs62078362) were in strong-LD with three other Th1- to Th2-associated SNPs in the *17q21.2* region (rs11871801, rs2006141, and rs4793090). Overall, these results highlight strong statistical association between vitamin D-binding sites and multiple genotypic variations linked to asthma, autoimmune, and allergic diseases.

Two SNPs in *ZPBP2* and *ORMDL3* (rs12936231 and rs4065275, respectively) that were previously shown to be functionally relevant to asthma (*Kelly et al., 2019*; *Schmiedel et al., 2016*; *Verlaan et al., 2009*), were also in high-LD with the four SNPs that we identified in the *VDR*-binding regions of

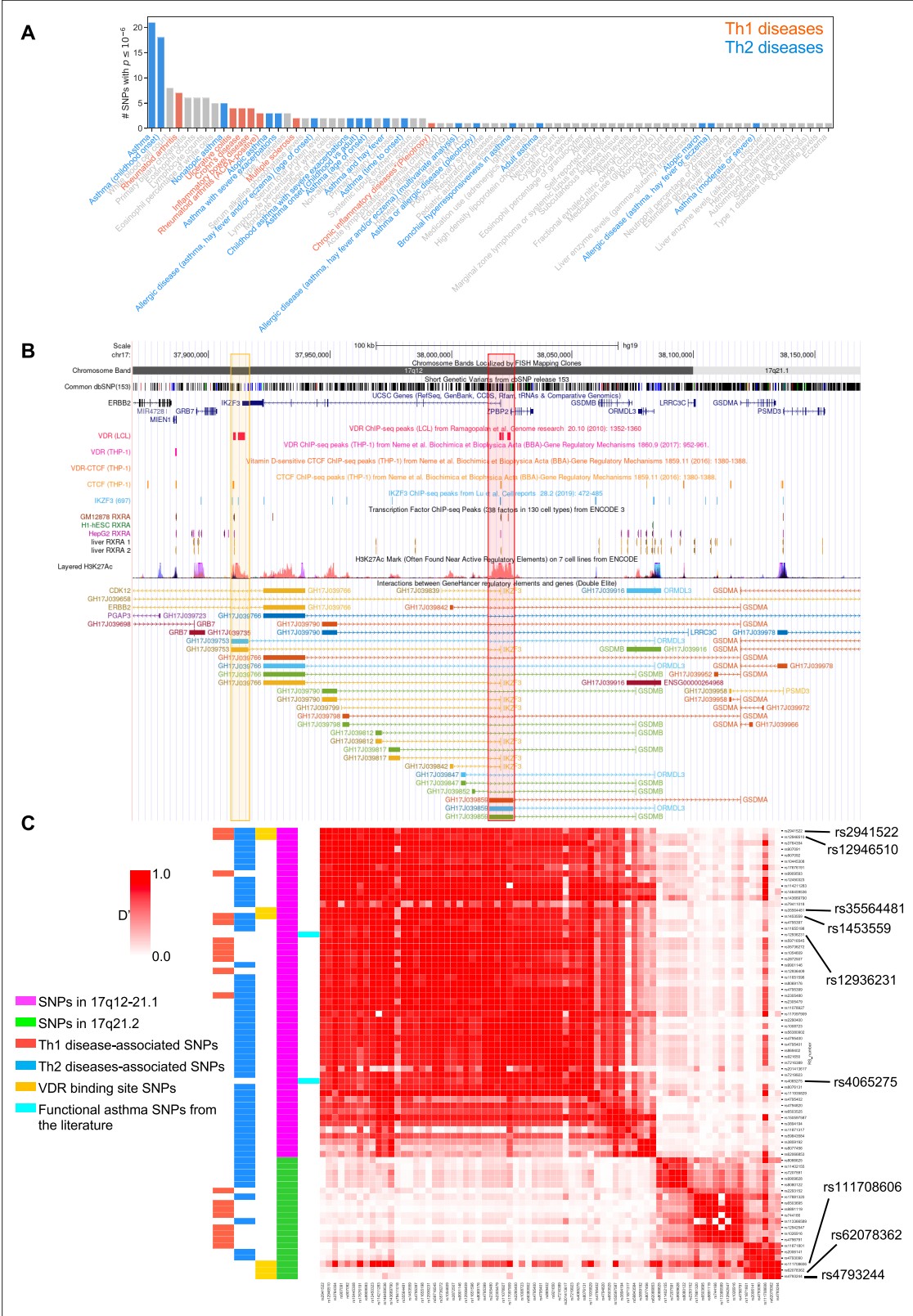

**Figure 1.** VDR-binding sites on Chr17q12 and q21.1 overlap with open chromatin signatures. (**A**) Significant disease and trait associations in 17q12-21.1 (*y*-axis), retrieved from the NHGRI EBI GWAS Catalog, with at least one significant single-nucleotide polymorphism (SNP) (*x*-axis) in Chr17q12-21.1. Colored bars denote autoimmune diseases mapped to Th1 (red bars)- or Th2 (blue bars)-driven immunity. (**B**) University of California Santa Cruz (UCSC) Genome Browser tracks showing chromosomal location, common dbSNPs, VDR-, RXR-, CTCF-, and IKZF3-binding sites and H3K27Ac marks present in

*Figure 1 continued*

the Chr17q12-21.1 locus. VDR-binding sites overlapping with active regulatory elements are highlighted by colored boxes. (**C**) Linkage disequilibrium (LD) between the significant SNPs in the Chr17q12-21.1 (magenta) and 21.2 (green) region, highlighting Th1/Th2-associated SNPs (red and blue, respectively), expression quantitative trait loci (eQTLs) in the VDR-binding regions in Chr17q12 21.1 and 21.2 (yellow), and functional asthma SNPs from the literature (cyan).

The online version of this article includes the following source data and figure supplement(s) for figure 1:

**Figure supplement 1.** GWAS disease associations and VDR binding sites of the 17q21.2 locus; overview of the VDAART analyses.

**Figure supplement 1—source data 1.** Results of the logistic regression analysis represented schematically in *Figure 1—figure supplement 1C*.

the *17q12-21.1* loci (*Figure 1C*). Importantly, in both the *17q12-21.1* and *17q21.2* regions, the VDR-binding sites coincide with *CTCF*- and *IKZF3*-binding sites, as well as H3K27ac peaks indicating active enhancer sites (*Figure 1B*, *Figure 1—figure supplement 1B*). In particular, the two enhancer sites in *17q12-21.1* (GH17J039753 and GH17J039859) were both predicted to interact with the *ORMDL3* promoter region (*Figure 1B*), and the enhancer site in *17q21.2* (GH17J042576) was predicted to interact with the *STAT5A* promoter region (*Figure 1—figure supplement 1B*). This suggests a similar mechanism to the one described in *Schmiedel et al., 2016*, whereby SNPs in LD with each other in these enhancer sites affect *VDR* and *CTCF* binding, the co-dependence of which in turn could modulate *ORMDL3* and *STAT5A* expression. Although it has been shown that *CTCF*-binding sites can be vitamin D sensitive (*Devereux et al., 2007*), no such sites were found in our genomic regions of interest (*Figure 1B*, *Figure 1—figure supplement 1B*).

Next, we investigated cord blood (CB) gene expression data in the VDAART (see Methods) and looked at the genes of interest in the *17q12* and *17q21* regions and their interaction with CB vitamin D levels as predictors of asthma/wheeze risk at age 3 years of age. After adjusting for relevant covariates (see Methods), the baseline expression term of *IKZF3* was not found to be significantly associated with asthma risk, while the interaction between vitamin D and *IKZF3* was statistically significant (p-val. <0.05), suggesting that the association of *IKZF3* expression with asthma risk was mediated by vitamin D levels in the CB (*Figure 1—figure supplement 1C*, *Figure 1—figure supplement 1—source data 1*).

Based on our bioinformatic analysis of the *Chr17q12* and *17q21* regions, we hypothesize that low vitamin D tissue levels are associated with decreased *IKFZ3* expression and increased expression of *ORMDL3*, *STAT3*, *STAT5A*, *STAT5B*, and *IL2*.

## *Vdr* is expressed in Th2 cells in allergic airway inflammation

To test our hypothesis of vitamin D activity at the Chr17q12-21 locus we utilized a mouse model of allergic airway inflammation and identified vitamin D responsive leukocyte subsets in house dust mite (HDM)-sensitized and challenged mice using flow cytometry (*Figure 2A*, *Figure 2—figure supplement 1A–C*). *Vdr* expression was significantly induced in HDM sensitized and challenged mice and almost absent in leukocytes isolated from control (vehicle) lungs (*Figure 2—figure supplement 2C*). Within the CD45+ population, *Vdr* expression was high in CD4+ T cells (p = 0.0357), while only very few CD8+ T cells and CD19+ B cells expressed *Vdr* (*Figure 2B*, *Figure 2—figure supplement 1B* HDM). Within the CD4+ T cell population, highest *Vdr* expression was detected in Gata3+ Th2 cells (p = 0.002 vs Tbet+ and p < 0.0001 vs Foxp3+ T cells). *Vdr* expression was very low in Tbet+ Th1 and CD4+Foxp3+ Treg cells (*Figure 2—figure supplement 1C* HDM). The differential expression of *Vdr* in Th2 cells was further validated by Western blot analysis using naive CD4+ T cells immediately after isolation (0 hr), after activation with CD3/CD28 for 48 hr and in in vitro polarized Th1, Th2, and iTreg cells with well-defined culture conditions (*Figure 2—figure supplement 2A*, *Figure 2—figure supplement 2—source data 1* and see Materials and methods). *Vdr* expression appeared as early as day 1 of Th2 culture conditions and gradually increased during the polarization process (*Figure 2—figure supplement 2B*, *Figure 2—figure supplement 2—source data 2*). In the absence of the *Vdr* ligand vitamin D, *Vdr* expression localized to the cytoplasm of Th2 cells. Addition of the *Vdr* ligand calcitriol led to a strong translocation of *Vdr* into the nucleus, suggesting functionality of the expressed receptor in the presence of its ligand (*Figure 2—figure supplement 2C*, *Figure 2—figure supplement 2—source data 3 and 4*). These results identify Th2 cells as vitamin D targets in allergic airway inflammation.

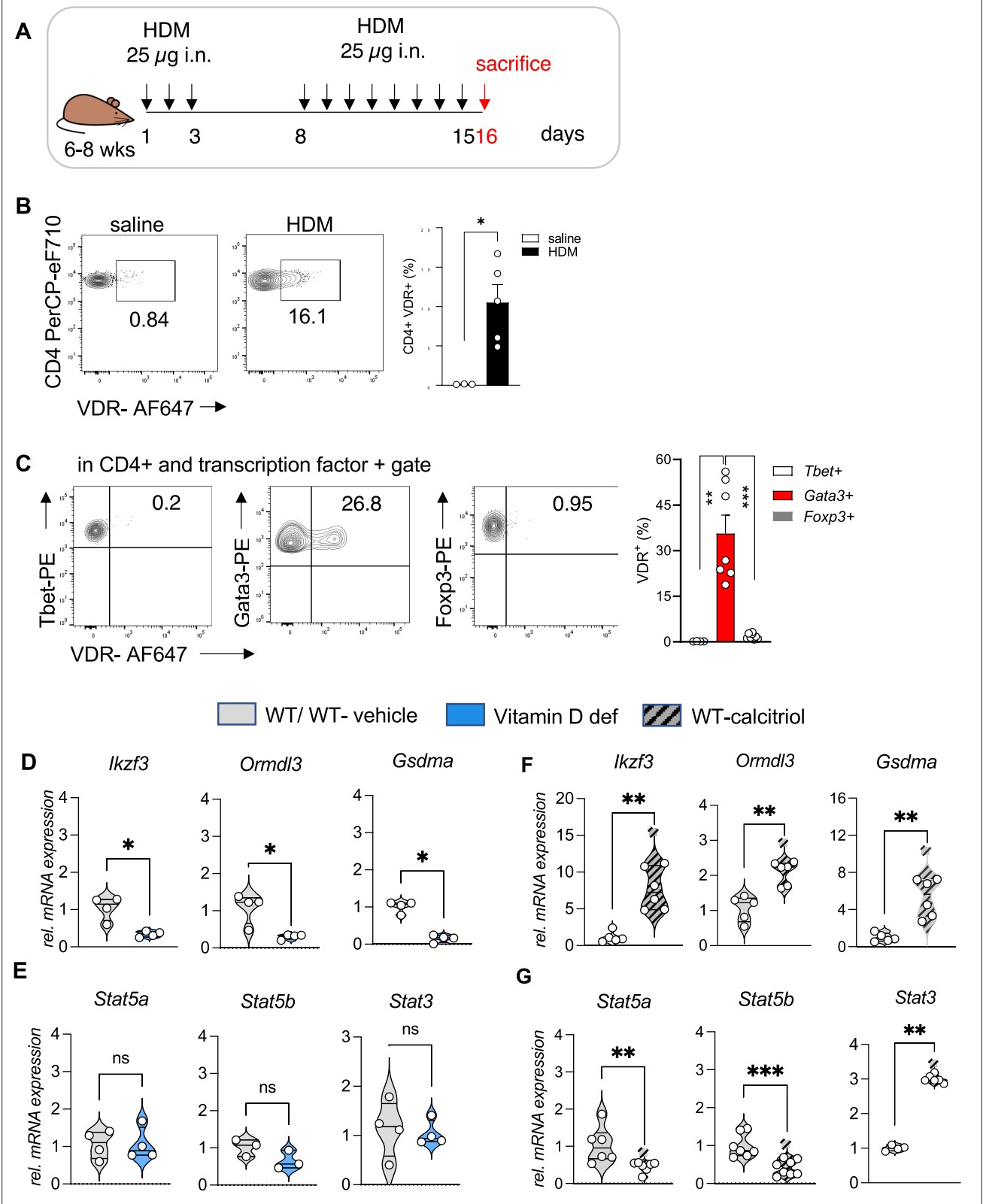

**Figure 2.** *Vdr* expression is elevated in Th2 cells and vitamin D regulated expression of genes on Chr1q12 and Chr17q21. (**A**) Scheme of the house dust mite (HDM)-induced airway inflammation protocol. (**B**) Flow cytometric analysis and cell frequencies of *Vdr* expression in CD45+ CD4+ T cells in the respective groups (*n* = 3–5). (**C**) Flow cytometric analyses and cell frequencies of *Vdr* expression in Th1 (Tbet+), Th2 (Gata3+), and Treg (Foxp3+) cells in the respective groups (*n* = 4–8). Quantitative Reverse transcription-polymerase chain reaction (RT-PCR) analysis of relative mRNA expression levels of

*Figure 2 continued on next page*

Figure 2 continued

genes encoded on Chr17q12-21.1 in (**D**) Th2 cultures of wild-type (WT) and vitamin D-deficient mice and (**E**) control and calcitriol-stimulated cultures. Quantitative RT-PCR analysis of relative mRNA expression levels of genes encoded on Chr17q21.2 in (**F**) Th2 cultures of WT and vitamin D-deficient mice and (**G**) control and calcitriol-stimulated cultures. Gene expression levels were normalized to the house keeping gene L32 and are expressed relative to the expression level in WT (**D, F**) or WT-vehicle (**E, G**) Th2 cells ($n \geq 4$ per group). Each symbol represents one mouse. Numbers in flow plots indicate percentages. Error bars indicate the standard error of the mean (SEM). Statistical tests: two-tailed Mann–Whitney $U$-test (**B, C**). *$p < 0.05$, **$p < 0.01$, ***$p < 0.001$. Data summarize results from two independent experiments.

The online version of this article includes the following source data and figure supplement(s) for figure 2:

**Figure supplement 1.** Baseline Vdr expression is confined to the CD4+ Foxp3 Teff cell population.

**Figure supplement 2.** *Vdr* expression is induced during CD4+ Th2 cell differentiation.

**Figure supplement 2—source data 1.** Original file for the Western blot analysis in *Figure 2—figure supplement 2A* (anti-Vdr, anti-b-actin).

**Figure supplement 2—source data 2.** Original file for the Western blot analysis in *Figure 2—figure supplement 2B* (anti-Vdr, anti-b-actin).

**Figure supplement 2—source data 3.** Original file for immunfluorescence images in *Figure 2—figure supplement 2C* stained for 4′,6-diamidino-2-phenylindole (DAPI), anti-CD4, anti-Vdr in in vitro polarized Th2 cells in the presence of calcitriol.

**Figure supplement 2—source data 4.** Original file for immunfluorescence images in *Figure 2—figure supplement 2C* stained for DAPI, anti-CD4, anti-Vdr in in vitro polarized Th2 cells in the absence of EtOH.

## Vitamin D status regulates the expression of key genes on Chromosome17q12-21.1 and Chr17q21.2

Taking cell-type-specific gene expression and selective expression of *Vdr* in Th2 cells into account, we next analyzed the expression of genes encoded on Chr17q12-21.1 and 17q21.2 loci in wild-type (WT) and vitamin D-deficient Th2 cultures as well as in Th2 cells post calcitriol stimulation.

Vitamin D deficiency differentially regulated the expression levels of several Chr17q12-21.1-encoded genes, with lower expression of *Ikzf3*, *Ormdl3*, and *Gsdma* (all $p = 0.0286$) (*Figure 2D*). Conversely, stimulation with calcitriol induced the expression of the same genes (all $p = 0.043$) (*Figure 2F*). Expression of the Chr17q21.2 genes *Stat3*, *Stat5a*, and *Stat5b* was not affected by vitamin D deficiency (*Figure 2E*). However, after calcitriol stimulation *Stat5a* ($p = 0.0082$) and *Stat5b* ($p = 0.003$) genes were strongly suppressed and *Stat3* ($p = 0.0048$) expression was increased in Th2 cells. (*Figure 2G*). These results hint to a protective role of vitamin D in allergic airway inflammation, by regulating the expression of the IL-2 downstream signaling molecules *Stat5a* and *Stat5b*.

## Deficiency in vitamin D signaling augments allergic airway inflammation

To decipher the protective role of vitamin D signaling in allergic airway inflammation, we employed vitamin D-deficient and $Vdr^{-/-}$ mice (*Figure 3A*, *Sakai et al., 2001*). HDM sensitization of vitamin D-deficient mice led to a stronger reaction to the allergen resulting in more prominent peri-bronchial and perivascular leukocytic infiltrates (*Figure 3B*, *Figure 3—source data 1*). This phenotype was accompanied by lower *Vdr* expression in CD4+ T cells isolated from HDM-exposed lungs compared to WT (*Figure 3C*). Vitamin D deficiency led to a robust increase in total immunoglobulin (Ig)E levels (*Figure 3D*). Histological findings were associated with higher total BAL leukocyte ($p = 0.0008$) (*Figure 3E*), eosinophil ($p = 0.0002$) (*Figure 3F*), and lymphocyte ($p = 0.0023$) (*Figure 3G*) numbers in vitamin D-deficient mice, while neutrophil numbers were not affected (*Figure 3—figure supplement 1A*). Th2 cell numbers ($p = 0.0127$) were also higher in the HDM-exposed mice (*Figure 3H*) in a manner that was independent of T helper cell skewing (*Figure 3—figure supplement 1B, C*). Differentiation and recruitment of Th17 and IL-10 producing CD4+ T cells was not affected by vitamin D deficiency (*Figure 3—figure supplement 1A, B*). To validate these findings from vitamin D deficiency, we next employed $Vdr^{-/-}$ mice (see Methods). As with the vitamin D-deficient mice, HDM sensitization and challenge of $Vdr^{-/-}$ mice led to an increased Th2 immune response. The lung phenotype in the $Vdr^{-/-}$ mice was marked by dense leukocytic infiltrates around and mucus plugs in the airways (*Figure 3B*). The augmented systemic and local immune response increased total IgE titers ($p = 0.0489$), and total BAL ($p < 0.0001$ vs WT HDM and $p = 0.003$ vs Vit D-deficient HDM), eosinophil ($p < 0.0001$ vs WT HDM and Vit D-deficient HDM), total lymphocytes ($p = 0.0194$ vs WT HDM and $p = 0.0421$ vs Vit D-deficient HDM), and Th2 cells ($p = 0.0009$ vs WT HDM) (*Figure 3D–H*). $Vdr^{-/-}$ did not significantly affect recruitment of neutrophils, Treg, and Th17 cells (*Figure 3—figure supplement 1A-C*).

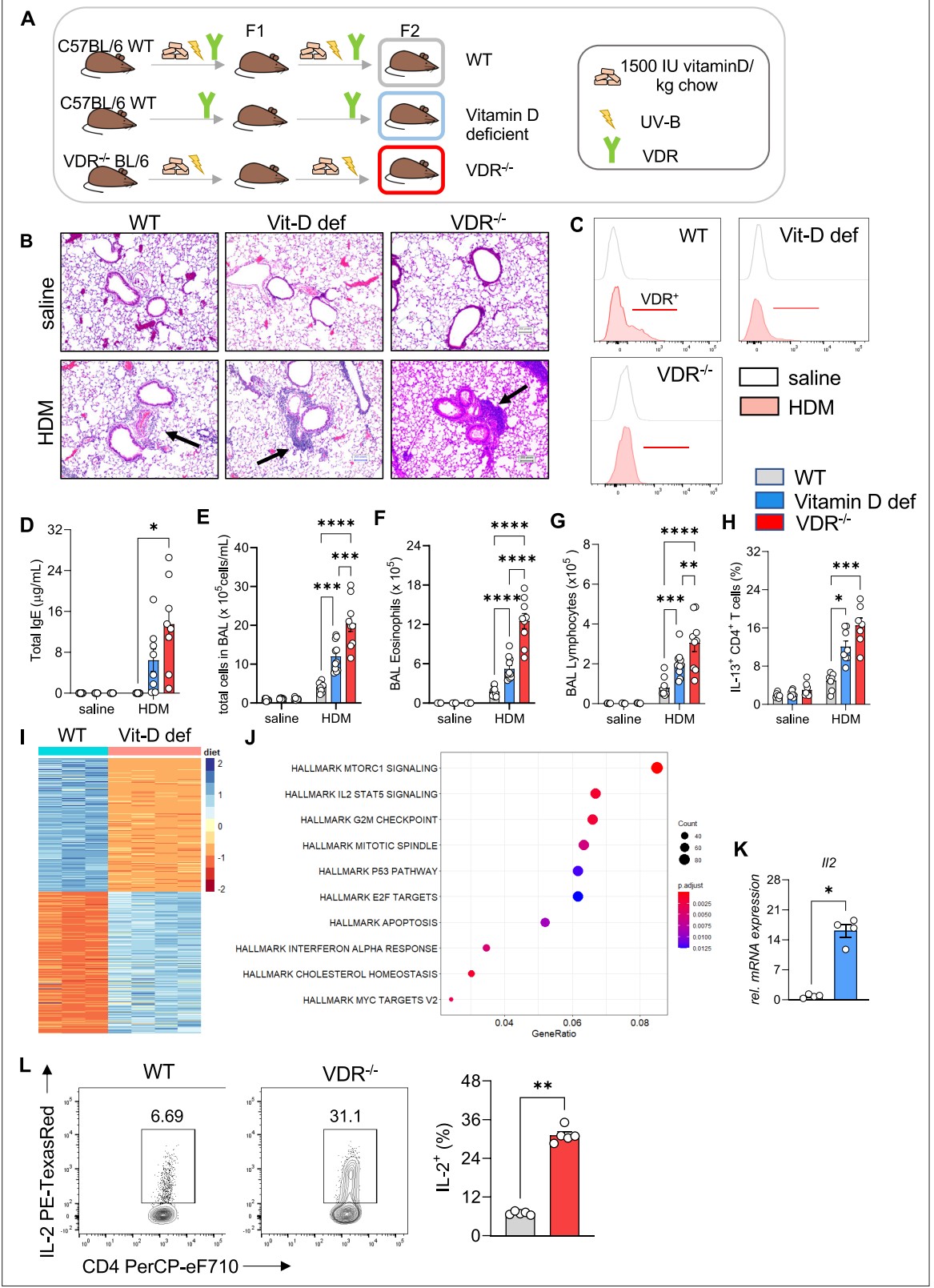

**Figure 3.** Vitamin D deficiency augments allergic airway inflammation. (**A**) Schematic presentation of the different mouse strains used. (**B**) Representative H&E-stained lung sections from saline and house dust mite (HDM)-exposed wild-type (C57/BL6; WT), vitamin D-deficient and *Vdr⁻/⁻* mice (×10 magnification). Black arrows point to the peri bronchial infiltrate. (**C**) Representative flow plots assessing *Vdr* expression in lung CD4+ T cells from the indicated groups. (**D**) Total IgE levels detected in the sera of respective groups. Absolute numbers of (**E**) leukocytes, (**F**) eosinophils, and

*Figure 3 continued on next page*

*Figure 3 continued*

(**G**) lymphocytes in the airways of the respective groups. Data summarize results from two or three independent experiments with *n* ≥ 8 per group. (**H**) Frequencies of IL-13 producing Th2 in the lung. Data summarize results from two or three independent experiments with *n* > 7 per group. (**I**) Heat map showing differentially expressed genes between WT and vitamin D-deficient Th2 cultures identified by RNA-sequencing analysis. Analysis and visualization were performed with DESeq2. (**J**) Overrepresentation analysis of Hallmark gene sets in the differentially expression gene set. The *x*-axis represents the gene ratio in the respective gene set. Dot sizes denote the number of genes in the respective pathway. The dot color indicates the false discovery rate (FDR)-adjusted p-value. Analysis and visualization were performed with the clusterProfiler package. (**K**) Quantitative RT-PCR analysis of IL-2 mRNA expression in Th2 cultures from the respective groups (*n* = 4 per group). Gene expression levels were normalized to the house keeping gene L32 and are expressed as fold-induction compared to WT Th2 cells. (L) Flow cytometric analysis and cell frequencies of IL-2 production in WT and *Vdr*⁻/⁻ Th2 cultures (*n* = 5 per group). Each symbol represents one mouse. Error bars indicate the standard error of the mean (SEM). Statistical significance was determined with: Mixed-effect analysis with Holm–Šidák's post hoc analysis (D–H) two-tailed Mann–Whitney *U*-test (**K, L**). *p < 0.05, **p < 0.01, ***p < 0.001, ****p < 0.0001. Data summarize results from two or three independent experiments with *n* > 8 per group.

The online version of this article includes the following source data and figure supplement(s) for figure 3:

**Source data 1.** Original images of H&E-stained lung section shown in *Figure 3B*.

**Source data 2.** Gene set enrichment analysis shown in *Figure 3J*.

**Source data 3.** Gene set enrichment analysis shown in *Figure 3J* (Hallmark gene sets).

**Figure supplement 1.** Vitamin D deficiency does not alter CD4+ IL-10 and IL-17A expression allergic airway inflammation.

**Figure supplement 2.** Vitamin D deficiency increases IL-13 expression in Th2 cells.

To explore the mechanisms underlying the augmented Th2 phenotype in the lungs of vitamin D-deficient and *Vdr*⁻/⁻ mice after HDM allergen challenge, we analyzed the transcriptional profiles of in vitro polarized Th2 cells from WT and vitamin D-deficient mice (*Figure 3I*). Differential expression and enrichment analysis revealed dysregulation of several pathways implicated in Th2 cell activation, cytokine production, proliferation, and survival with prominent changes in the IL-2/STAT5 pathway in vitamin D-deficient Th2 cells (p = 0.0014) (*Figure 3J*, *Figure 3—source data 2 and 3*). The effects of impaired vitamin D signaling on these pathways were confirmed by qRT-PCR and flow cytometry. Baseline expression of IL-2 was markedly increased at both the RNA (p = 0.0286) and protein levels (p = 0.0079) in vitamin D-deficient and *Vdr*⁻/⁻ Th2 cells (*Figure 3K, L*). In vitamin D-deficient and *Vdr*⁻/⁻ Th2 cell in vitro incubation, the elevated IL-2 levels were accompanied by a sharp increase in IL-13 production (VitD deficient: p < 0.0001; *Vdr*⁻/⁻: p = 0.0141) (*Figure 3—figure supplement 2*).

Taken together, these results indicate a critical role for vitamin D/*Vdr* signaling in regulating type 2 inflammation and identify the IL-2/Stat5 pathway as a downstream target of vitamin D in Th2 cells.

## Vitamin D suppresses the activation of the IL-2/Stat5 pathway and cytokine production in Th2 cells

To ascertain the vitamin D-dependent regulation of the IL-2/Stat5 pathway and the impact on the effector program of Th2 cells, we first analyzed the transcriptional profile of WT Th2 cells exposed to calcitriol during differentiation (*Figure 4A*). Gene ontology enrichment analysis of biological processes revealed regulation of several processes impacting inflammatory responses, including chemotaxis and activation of cells (*Figure 4—source data 1*). Calcitriol stimulation of Th2 cells affected several immune-related disease pathways, including asthma (*Figure 4—figure supplement 1*, *Figure 4—source data 2*). Gene set enrichment analysis (GSEA), performed on the differentially expressed genes, highlighted the negative regulation of IL-2/Stat5 pathway in Th2 cells by calcitriol (*Figure 4B*, *Figure 4—source data 3*). Calcitriol stimulation suppressed the expression of *Il2* (p < 0.0001), *Stat5a* (p < 0.0008), and *Stat5b* (p < 0.0008) genes (*Figure 4C*). Concordantly, levels of activated Itk (phospho-Itk) (p = 0.029) and expression of the IL2Rβ (CD25) (p = 0.0441) were reduced on a per cell basis (*Figure 4D*).

Next, the impact of calcitriol stimulation on the effector function of Th2 cells was assessed by flow cytometric analysis for the T2 cytokine IL-13. Activation of the vitamin D/*Vdr* pathway led to significant suppression of IL-13 production in CD4+ T cells (p = 0.0012) (*Figure 4E*).

To further delineate the molecular impact of vitamin D on Th2 cell biology, we integrated the murine transcriptional profile with a recently developed human Protein–Protein Interaction (PPI) database (*Silverbush and Sharan, 2019*). By modeling the signal transduction process as a sequential path on the PPI network that starts from a receptor and propagates downstream to a transcription

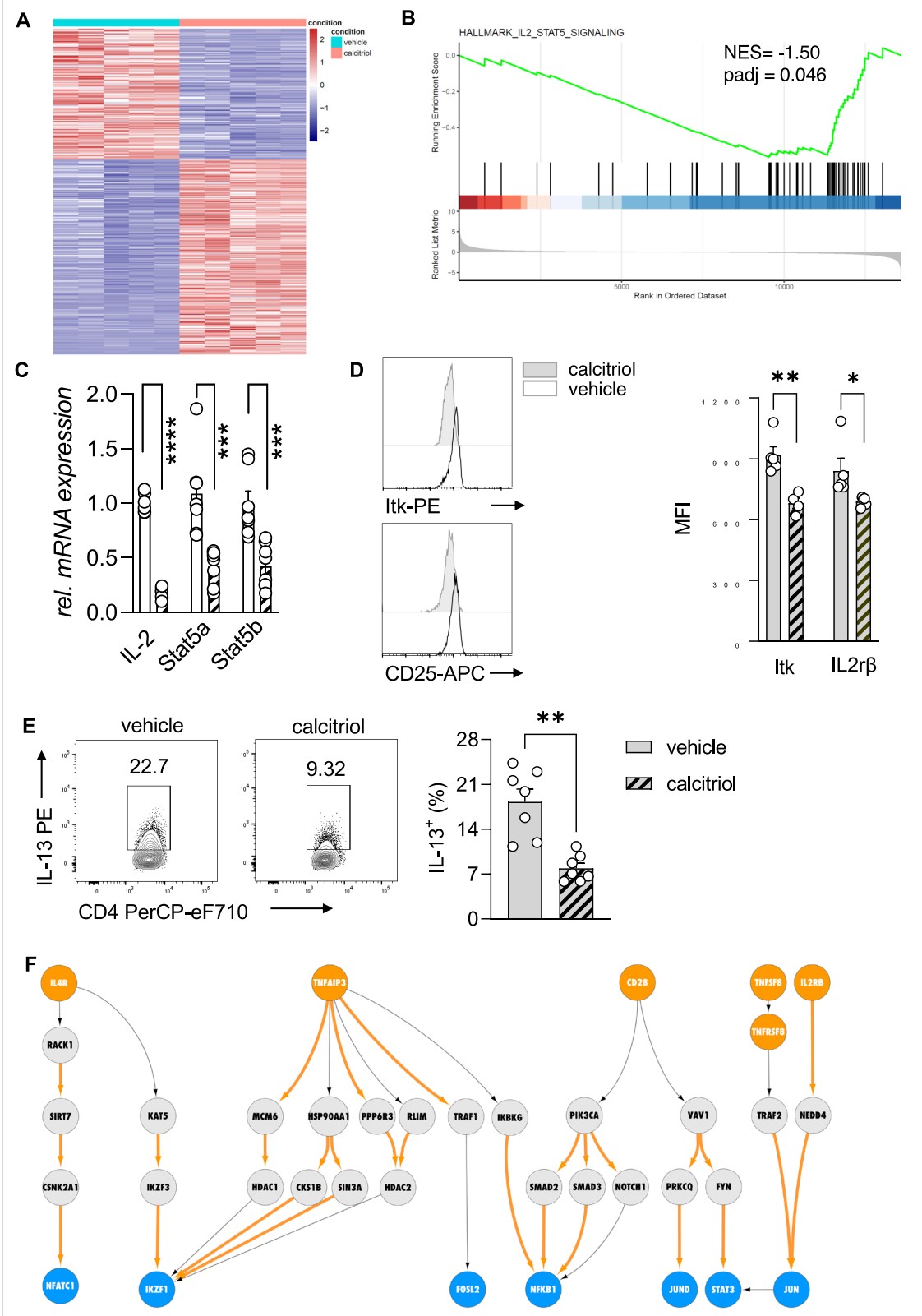

**Figure 4.** Vitamin D stimulation of polarizing Th2 cells suppresses the IL-2/Stat5 pathway. (**A**) Heat map of differentially expressed genes between vehicle (EtOH [% vol/vol])- and calcitriol-treated wild-type (WT) Th2 cells. Analysis and visualization were performed with DESeq2. (**B**) Gene set enrichment analysis (GSEA) of differentially expressed genes. Analysis and visualization were performed with the clusterProfiler package. (**C**) Quantitative RT-PCR for Il-2, Stat5a, and Stat5b mRNA expression in Th2 cells exposed to either vehicle or calcitriol. Gene expression levels were normalized to the

*Figure 4 continued on next page*

*Figure 4 continued*

house keeping gene L32 and are expressed as fold-induction compared to control cells. (**D**) Flow cytometric analysis and mean fluorescence intensities of Itk and Il2rβ expression in indicated groups. (**E**) Flow cytometric analysis and quantification of IL-13 expression in respective groups. (**F**) Network analysis to visualize vitamin D-induced molecular changes in the molecular interactome of calcitriol-stimulated CD4+ Th2. Predefined set of cell membrane receptors were set as starting points (orange nodes) and annotated transcription factors were set as ending points (blue). All intermediate proteins on the Protein–Protein Interaction (PPI) are visualized as gray nodes. Molecular interactions only overrepresented in calcitriol-stimulated cells are shown as orange arrows. Each symbol in the bar graphs represents one individual sample and data summarize the results from two or three independent experiments with $n \geq 4$. Error bars indicate the standard error of the mean (SEM). Statistical test: two-tailed Student's *t*-test (**C, E**) and two-tailed Mann–Whitney *U*-test (**D**). *p < 0.05, **p < 0.01, ***p < 0.001, ****p < 0.001.

The online version of this article includes the following source data and figure supplement(s) for figure 4:

**Source data 1.** Gene ontology enrichment analysis for *Figure 4*.

**Source data 2.** Gene ontology enrichment analysis for *Figure 4* (Hallmark gene sets).

**Source data 3.** Gene set enrichment analysis for *Figure 4*.

**Source data 4.** The subset of receptors and transcription factors selected as input for our network analysis.

**Source data 5.** Protein names for the subset of receptors and transcription factors selected as input for our network analysis.

**Figure supplement 1.** KEGG (Kyoto Encyclopedia of Genes and Genomes) pathways analysis post vitamin D stimulation.

**Figure supplement 2.** Active signaling pathways in Th2 cells.

factor via multiple intermediate proteins (for details see Materials and methods, *Figure 4—figure supplement 2A*), we inferred the active signaling flow in Th2 cells for control (vehicle) and calcitriol-stimulated Th2 cells. As input of our network analysis, we selected a subset of receptors and transcription factors that are known to be involved in CD4+ T cell activation, Th2 cell differentiation, and cytokine production (*Figure 4—source data 4*). For clarity of the figure, we display the gene names. Corresponding protein names are attached in *Figure 4—source data 5*. In control Th2 cells, our network analysis revealed the engagement of a variety of receptors including IL4R, IL2RA, CD28, and TNF receptors. Downstream to these receptors, the most active signaling paths highlighted the activation of STAT3, STAT5A/B, NFkB1, and IKZF1 transcription factors (*Figure 4—figure supplement 2B*) via multiple adapter proteins of the TRAF and JAK family. Interestingly, these transcription factors are known to be involved in differentiation and activation pathways in Th2 cells. Conversely, after calcitriol stimulation, signal from IL4R, CD28, and TNFSF8 propagated over different interaction partners (*Figure 4F*). Calcitriol-specific edges are highlighted in orange color. Molecular mediators included the cytoskeletal proteins VAV1 and RACK1 and emphasized activation of the transcription factors AP-1 (FOS/JUND dimer) and NFATC1.

Our network analysis highlighted a central role of the transcriptional repressor IKZF1 (IKAROS) upon calcitriol stimulation, which was targeted by multiple downstream mediators including IKZF3 (AIOLOS). Notably, the IKZF3 gene encodes for the AIOLOS protein, a direct interaction partner of the IKZF1-encoded IKAROS. Both transcription factors, as homo- and heterodimers, suppress IL-2 expression on the transcriptional levels in mouse and human lymphocytes (*Quintana et al., 2012*). Calcitriol increased the number of edges connecting intermediate proteins to the transcriptional repressor IKAROS. Among the connecting proteins, AIOLOS interaction with IKAROS was implicated in repression of IL-2 expression in Th2 cells by vitamin D.

Absence of STAT5A/B mediators in the active signaling paths of calcitriol-stimulated Th2 cells further suggested repression of the Th2 cell STAT5 pathway with calcitriol compared to control (*Figure 4—figure supplement 2B*). To ensure that the above findings were not restricted to the C57BL/6 mouse strain, the inverse experiment was performed in Balb/c mice. This mouse strain is commonly used for type 2-driven inflammation.

Taken together, these results indicate that the specific suppression of the IL-2/Stat5 pathway by vitamin D results in reduced cytokine production by Th2 cells. Our findings suggest that this suppression was mediated by the vitamin D-dependent regulation of IKAROS (IKZF1) through the induction of AIOLOS (IKZF3).

## Vitamin D supplementation alleviates the allergic phenotype in the lung by suppressing type 2 cytokine production in a dose-dependent manner

To test if vitamin D supplementation prevents the development of T2-driven allergic inflammation in the lung, custom rodent diets with select vitamin D doses were used: 400 IU/kg chow (400 IU; low), 1000 IU/kg chow (1000 IU; regular), and 4000 IU/kg chow (4000 IU; high) (*Figure 5—figure supplement 1A* see Materials and methods). The 1000 IU group was used as the control group, as it contains the average amount of vitamin D fortified in standard rodent chow. HDM exposure led to recruitment of leukocytes into the lung tissue and BAL (*Figure 5A, B*; *Figure 5—figure supplement 1B*, *Figure 5—source data 1*), with a predominant eosinophilic and lymphocytic infiltrate as well as high numbers of IL-13+ Th2 cells (*Figure 5C–E*). There were trends for higher cellular infiltration in the airways when mice were supplemented with the low vitamin D doses (400 IU), but these inflammatory responses were not significantly different from the phenotype observed with 1000 IU. Lung and airway pathology were similar between these dosing groups, with similar numbers and frequencies of the different leukocyte populations. In sharp contrast, dietary supplementation with the higher vitamin D dose (4000 IU) decreased the inflammatory phenotype in HDM-exposed mice compared to the 400 and 1000 IU dosed animals. Concordant with the histological findings, BAL leukocytes (p < 0.0001) were decreased with marked reductions in BAL eosinophil (p = 0.0016), lymphocyte (p = 0.026), and neutrophil numbers (p = 0.0005) (*Figure 5B, D*; *Figure 5—figure supplement 1C*), and the frequency of IL-13+ Th2 cells was significantly reduced in the lungs with 4000 IU of vitamin D (p = 0.0066 vs 400 IU HDM; p = 0.0011 vs 1000 IU HDM) (*Figure 5E*). Of note, dietary supplementation with high vitamin D levels, increased *Vdr* expression in CD4+ T cells from HDM-exposed lungs (p < 0.0001) (*Figure 5—figure supplement 1C*).

To ascertain that dietary supplementation with vitamin D specifically altered mouse Th2 cell differentiation and the effector program, splenic naive CD4+ T cells from 400 and 4000 IU mice were studied. Of interest, naive CD4+ T cells did not express *Vdr* until they were exposed to Th2-polarizing conditions. Gata3 expression on per cell basis was lower in 4000 IU Th2 cells (p = 0.0286) (*Figure 5—figure supplement 2A*). This was accompanied by higher *Vdr* protein expression (p = 0.0286) (*Figure 5—figure supplement 2B*). The expression of IL-2 (p = 0.0004) and the Th2 cytokines IL-13 (p = 0.0159) and IL-5 (p = 0.0159) were lower in 4000 IU Th2 cultures (*Figure 5—figure supplement 2C–E*).

To explore the specific induction of Aiolos expression for vitamin D-mediated suppression of IL-2 production in Th2 cells, Aiolos expression was monitored throughout Th2 differentiation. Aiolos expression gradually increased under Th2-polarizing conditions in CD4+ T cells, starting as early as day 1 of in vitro culture (*Figure 5F*). Simultaneous exposure to calcitriol further increased Aiolos protein levels compared to vehicle in Th2 cells (p = 0.0079) (*Figure 5G*).

To test if vitamin D exerts its inhibitory effect on IL-2 production via Aiolos induction, *Ikzf3*$^{-/-}$ mice were employed, exposed to Th2-polarizing conditions and assessed for IL-2 production after calcitriol exposure. IL-2 production in control *Ikzf3*$^{-/-}$ Th2 cultures was significantly higher compared to WT Th2 cells. Calcitriol significantly reduced IL-2 levels in both, WT (p = 0.0001) and *Ikzf3*$^{-/-}$ Th2 cells (p = 0.0001) (*Figure 5H*); however, the suppressive effect of calcitriol on IL-2 production was significantly higher in WT Th2 cells (~65% inhibition) compared to *Ikzf3*$^{-/-}$ Th2 cells (~40% inhibition) (p = 0.0001) (*Figure 5H*).

Taken together, these results indicated that vitamin D can regulate Aiolos expression and suggested a role for Aiolos as a downstream effector for selectively fine-tuning Th2 cell IL-2 production by vitamin D.

## Discussion

We utilized human genetics to identify SNPs on chromosome 17q12-21.1 and 17q21.2 that were within GWAS regions for Th1 and Th2 autoimmune diseases. Of note, many of these SNPs were within *VDR*-binding sites. We then determined by eQTL analysis of the genes in these two genomic regions that IKZF3 expression was controlled by four cis-eQTLs in the enhancer region targeting ORMDL3 and IKZF3, and in the enhancer region targeting ORMDL3, GSDMA, and GSDMB. Next, we demonstrated that two SNPs in CTCF-binding sites were in strong LD with these four eQTL SNPs and that the two enhancer sites in 17q12-21.1 were both predicted to interact with the ORMDL3 promoter region,

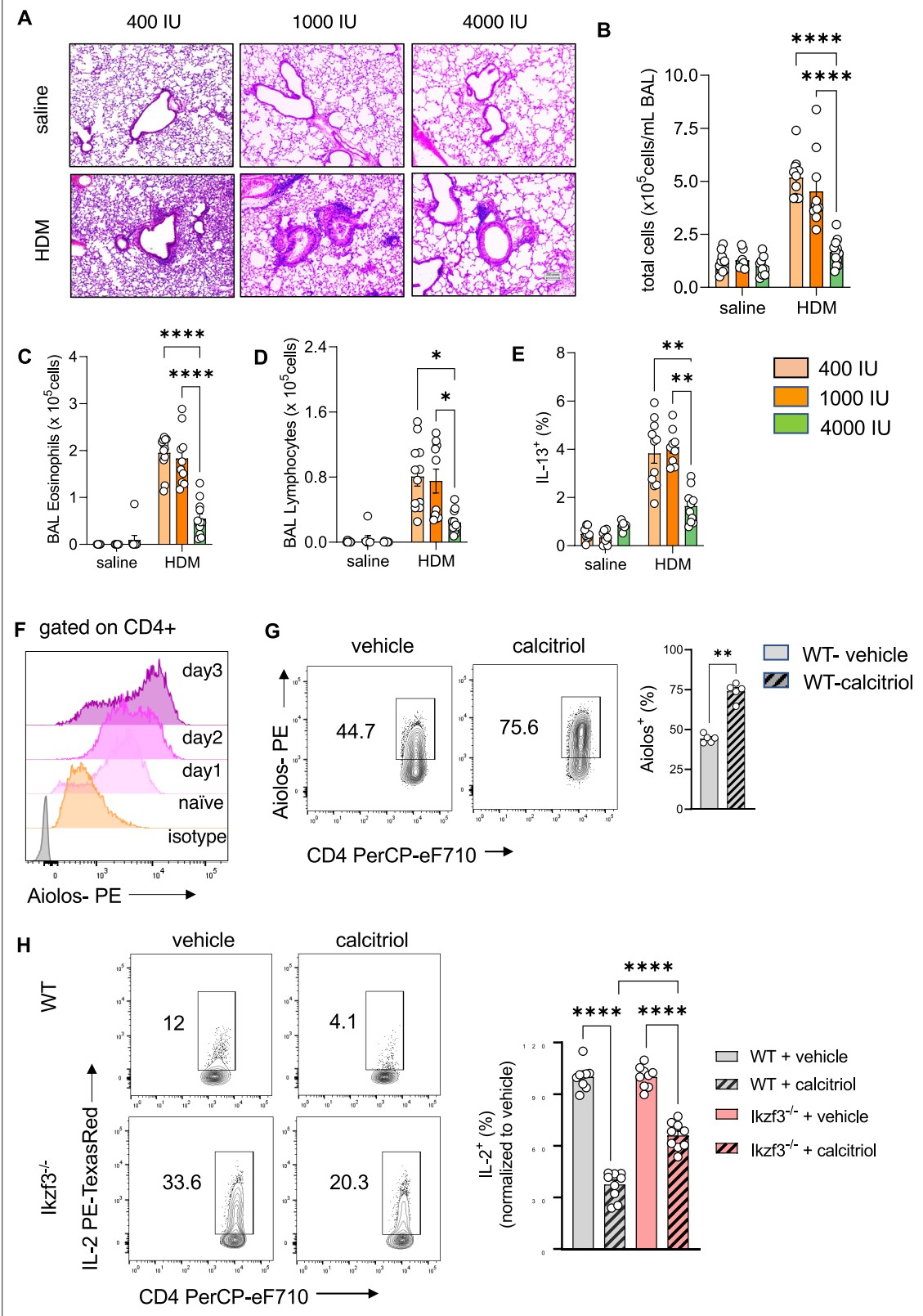

**Figure 5.** Prenatal vitamin D supplementation protects from asthma development. (**A**) H&E staining of representative lung sections of indicated experimental groups. Arrowheads point to peri bronchial infiltrates. Absolute numbers of total BAL (**B**) leukocytes, (**C**) eosinophils, (**D**) lymphocytes, and (**E**) lung IL-13+ Th2 cells (n ≥ 8 per group). (**F**) Flow cytometric analysis of Aiolos expression in differentiating Th2 cells at indicated time points. (**G**) Flow cytometric analysis of Aiolos expression in control and calcitriol stimulated Th2 cells. (**H**) Flow cytometric analysis and frequencies of IL-2+ cells in wild-

*Figure 5 continued on next page*

*Figure 5 continued*

type (WT) and *Ikzf3*−/− cultures post indicated treatment. Each symbol represents and independent sample (*n* ≥ 9 per group). Flow cytometric analysis and quantification of *Vdr* expression in lung CD4+ T cells of the respective groups. Each symbol represents one individual mouse and data summarize the results from two to three independent experiments with *n* > 8 per group. Error bars indicate the standard error of the mean (SEM). Statistical significance was determined with mixed-effect analysis with Holm–Šidák's post hoc analysis (**B–E**), two-tailed Mann–Whitney *U*-test (**G**) and one-way analysis of variance (ANOVA) with Holm–Šidák's post hoc analysis (**H**). *$p < 0.05$, **$p < 0.01$, ****$p < 0.0001$.

The online version of this article includes the following source data and figure supplement(s) for figure 5:

**Source data 1.** Original images of H&E-stained lung section shown in *Figure 5A*.

**Figure supplement 1.** Prenatal vitamin D supplementation increases *Vdr* expression in CD4+ T cells.

**Figure supplement 2.** Prenatal vitamin D supplementation mitigates Th2 development and cytokine production in vitro.

**Figure supplement 3.** Schematic summary of vitamin D molecular genetics in the 17q12-21 region.

and the enhancer site in 17q21.2 was predicted to interact with the STAT5A promoter region thus, suggesting a plausible human molecular genetic mechanism by which vitamin D could activate IKZF3 to repress ORMDL3 and immune system development. Finally, we leveraged bulk RNAseq in the CB of the VDAART trial to show that an interaction for vitamin D level and IKZF3 expression that was associated with reduced asthma risk at 3 years of age. *Figure 5—figure supplement 3* depicts the influence of vitamin D-binding sites on the genes in the 17q12-21 region.

To confirm our human molecular genetic findings, we chose to utilize mouse models of allergic lung inflammation. We show strong *Vdr* expression in CD4+ Th2 cells and alterations in chr17q12-21.1 and 17q21.2 gene expression induced by vitamin D status. Expression of *Vdr* has been reported for airway epithelial cells and different immune cell types present in the lung (*Pfeffer and Hawrylowicz, 2018*). In lymphocytes, ex vivo and in vitro studies describe low baseline *Vdr* transcript expression levels that were transiently increased by activating signals and calcitriol stimulation (*Baeke et al., 2010*). Here, we report a differential *Vdr* expression pattern in CD4+ T cell subsets from allergic inflamed lungs and post lineage-specific cytokine stimulation in vitro, with strongest expression in Th2 cells. Activation of *Vdr* suppressed IL-2, IL-5, and IL-13 production by Th2 cells. The expression of *Vdr* in Gata3+ Th2 cells in the lung might represent an endogenous control mechanism to curb the pathogenic cell activity and cytokine production after repetitive allergen contact. Ex vivo and in vitro Th1, nTreg, and iTreg cells expressed low *Vdr* protein levels at baseline. It remains to be determined whether other Th subsets described in allergic inflammation express *Vdr* at baseline and if either calcitriol or other secondary signals, specific for each subset, could induce *Vdr* expression during inflammation.

Vitamin D deficiency that had been introduced over two generations, impaired vitamin D signaling and led to an augmented allergic phenotype in the lung after allergen exposure with a dominating Th2 signature locally and systemically. This was likely due to specific deregulation of Th2 cells instead of Th cell skewing, since neither Th17 nor Treg development and recruitment were significantly affected. Similar findings regarding the effects of vitamin D in controlling Treg/IL-10 and dampening Th2 responses have been reported, for example, in *Taher et al., 2008* and in offspring of mice that had been subjected to vitamin D deficiency in the third trimester of their pregnancy (*Vasiliou et al., 2014*). In contrast, in mice supplemented with high vitamin D doses over two generations, *Vdr* expression and consequently vitamin D responsiveness was higher in CD4+ T cells from the animals' lungs, almost completely abrogating the inflammatory process in the lung after allergen challenge. Additional pathways, including the induction of IL-10 production by CD4+ T cells as well as a direct induction of Foxp3+ T reg cells could have further contributed to the observed protective effect of vitamin D supplementation (*Palmer et al., 2011*; *Kang et al., 2012*). While the vitamin D dose in the mice was chosen to replicate the VDAART trial dose in the mouse model, the duration of supplementation was adapted to the protocol for generating vitamin D-deficient mice. Both factors might influence the outcome here compared to prior rodent studies with inconclusive results. We provide a discussion of vitamin D toxicity in our response to the reviewers. More work to determine the exact levels of vitamin D that confer this protective effect in the mice would also be worthy of further investigation.

Differentiation of Th2 cells is driven by IL-4-induced Gata3 expression in naive CD4+ T cells. This process depends on a positive feedforward loop by IL-2 and its downstream STAT5 signaling pathway for the effective expression of IL-4 (*Zhu et al., 2003*). Therefore, either interfering with Gata3 expression or with the IL-2 pathway represents a promising strategy to fine-tune Th2-driven immunity. Our

data highlight the specific regulation of the IL-2/STAT5 pathway by vitamin D. Vitamin D deficiency and impaired *Vdr* signaling caused elevated IL-2 production by Th2 cells, leading to high IL-13 production. Elevated IL-13 levels in vivo, could amplify the local T2 responses in the lung. In sharp contrast, exposure of polarizing Th2 cells to calcitriol was able to suppress IL-2 production, reduce expression and activation of STAT5A/B and therefore reduce IL-13 production by Th2 cells. The transcription factor AIOLOS has previously been shown to suppress IL-2 expression at the transcriptional level (*Quintana et al., 2012*). We report here a vitamin D-mediated induction of Aiolos expression in Th2 cells, that is impaired by vitamin D deficiency. High Aiolos expression coincided with lower IL-2 production, ameliorated STAT5A/B expression, and reduced IL-13 production in Th2 cells. In vitro Th2 polarization of naive CD4+ T cells isolated from mice supplemented with higher vitamin D levels, without further exogenous calcitriol stimulation, developed a weakened Th2 phenotype, characterized by lower IL-2, IL-5, and IL-13 production. Since naive CD4+ T cells do not express *Vdr*, these results, together with our human data, implicate changes in chromosomal accessibility by high vitamin D in the developing immune system, which needs further investigation.

Experiments here with *Ikzf3⁻/⁻* mice demonstrated significant reductions in vitamin D-mediated control of Th2 cell cytokine production, but the vitamin D response was not completely impaired, suggesting additional mechanisms for vitamin D regulation of Th2 cells – several possibilities were uncovered by our PPI analyses. These will be the subject of future research. In addition, we only investigated a Th2 mouse model of asthma and have not extended our mouse work to Th1 models of autoimmunity. We acknowledge that the impact of vitamin D on Th2 biology is conflicting in the literature. While several groups report Th2 promoting activity, we, and others, show inhibition of type 2 cytokine production (*Boonstra et al., 2001*; *Cantorna et al., 2015*; *Pichler et al., 2002*; *Staeva-Vieira and Freedman, 2002*). These discrepancies could be due to the model system studied, for example, peripheral blood mononuclear cells (PBMC) and purified CD4+ T cells, or the dose of vitamin D or the mouse strain.

We have still to consider other asthma genes that might be linked to these two genetic loci as there are likely other *VDR*-binding sites of importance in both asthma and Th1 autoimmunity. Our preliminary results confirm that three asthma susceptibility loci: 2q12.1 (IL1RL1), 6p21.32 (HLA-DQA1/B1/A2/B2), and 22q12.3 (IL2RB) each have *VDR*- and IKZF3-binding sites either in enhancers predicted by GeneHancer to target these genes or within these genes themselves. It is important to consider other data that might have enhanced the results that we present here. For our human genetic data, additional use of ChIP or co-IP to establish STAT induction and activation would have been of potential value. For our mouse studies, additional cytokine measurements in the mice as well as measurement of airway resistance would have added to the pathophysiological data linking IKFZ3 expression to TH2 response. Finally, quantification of histology and confocal images could provide an overview of *VDR* expression in the lungs.

In summary, we have leveraged clinical trial data, molecular genetic bioinformatic data, and mouse model data to outline for the first time, a comprehensive molecular genetic mechanism for how vitamin D influences not only asthma development, but both Th1 and Th2 autoimmune disease. The significance of these findings relates to the high prevalence of vitamin D deficiency, especially during pregnancy, and the strong possibility that the epidemic of asthma and autoimmunity might be significantly reduced if serum vitamin D levels were elevated worldwide.

## Materials and methods
### In silico analysis of VDR-binding sites on Chr17q12-21

For our in silico analysis, we focused on two genomic regions that are known to harbor variants associated with asthma and autoimmune disease susceptibility. The first one is a 240-kb region that covers parts of the 17q12 and 17q21.1 loci and contains IKZF3, ZPBP2, ORMDL3, GSDMA, and GSDMB (chr17: 37,899,254–38,139,253; hg19) and the second one is a 420-kb region in the 17q21.2 locus that includes STAT3, STAT5A, and STAT5B (chr17: 40,330,000–40,750,000; hg19). When selecting these genomic ranges, our criterion was to make the window inclusive of any potential enhancer regions targeting the above genes of interest. We used the GeneHancer (*Fishilevich et al., 2017*) track in the UCSC Genome Browser (http://genome.ucsc.edu) to determine predicted enhancer–target gene interactions. We obtained VDR, IKZF3, CTCF, and vitamin D-sensitive CTCF-binding sites from *Neme*

*et al., 2016*; *Neme et al., 2017*; *Ramagopalan et al., 2010*; *Lu et al., 2019*, converted their genomic coordinates to hg19 if a different genome build was used in the original publication, and added them as custom tracks on the Genome Browser. We retrieved the known disease and trait associations in these loci from the NHGRI-EBI GWAS Catalog (downloaded on 7/26/2021) (*Buniello et al., 2019*) and retained all hits that were deemed significant (p < 10⁻⁵). We downloaded significant single-tissue eQTLs for IKZF3 and PSMCIP3 in all tissues from the Common Fund (CF) Genotype-Tissue Expression Project (GTEx) Analysis Release V8 (dbGaP Accession phs000424.v8.p2), through the GTEx Portal (https://www.gtexportal.org/home/) in July 2021. We calculated LD using the NCI LDmatrix Tool (https://ldlink.nci.nih.gov/?tab=ldmatrix), and by selecting all available populations.

## The VDAART

The design, eligibility, and initial results of the trial have been published (*Litonjua et al., 2014*; *Litonjua et al., 2016*; *Litonjua et al., 2020*; *Wolsk et al., 2017a*; *Wolsk et al., 2017b*; *Verlaan et al., 2009*; *Miller et al., 2014*). Pregnant women were recruited from 3 clinical sites across the United States – Boston Medical Center, Boston, MA; Washington University at St. Louis, St. Louis, MO; and Kaiser Permanente Southern California Region, San Diego, CA. The Data Coordinating Center was based in the Channing Division of Network Medicine, Brigham and Women's Hospital, Boston, MA. Eligible participants were women between the ages of 18 and 39 years, who presented between the estimated gestational ages of 10 and 18 weeks; who had a history of asthma, eczema, or allergic rhinitis, or whose partner (biologic father of the child) had a history of asthma, eczema, or allergic rhinitis; who was a non-smoker; and who was English- or Spanish-speaking, with intent to participate for 4 years (up to the third birthday of the child). The VDAART study protocol was approved by the institutional review boards at each participating institution and at the Brigham and Women's Hospital. All women provided a written informed consent. The trial was registered with clinical trials.gov: clinicaltrials.gov Identifier: NCT00920621.

## Gene expression analysis in VDAART cohort

To understand the putative mechanism linking IKZF3 expression and vitamin D to asthma risk we considered umbilical CB RNA-seq data from the VDAART trial. *Figure 1—figure supplement 1C* shows the study design. Total RNA was isolated from CB using the QIAGEN PAXgene Blood RNA Kit (QIAGEN) according to the manufacturer's protocol. Quality was assessed using the Nanodrop 8000 spectrophotometer (ThermoFisher Scientific). Sequencing libraries were constructed with the TruSeq Stranded Total RNA Library Prep Globin Kit (Illumina) and the NEXTflex Small RNA sequencing Kit v3 (Boo Scientific). Sequencing was performed using a NextSeq 550 instrument (Illumina). Trimmed reads were mapped to The GRCh38 reference genome using STAR (*Dobin et al., 2013*). Read counts were computed with htseq (*Anders et al., 2015*). The data include 443 expression profiles of pregnant women with and without asthma undergoing measurements of vitamin D levels in their CB. Each sample is matched with the observed patterns of wheezing and asthma in the newborn. We filtered all the samples where information about vitamin D levels, asthma/wheezing patterns, or other covariates were missing, obtaining 393 expression profiles. Data were normalized with the voom function of the edgeR R package (*McCarthy et al., 2012*), and vitamin D levels were binarized using 30 ng/ml as threshold. We performed logistic regression of asthma status as a function of the expression levels of IKZF1 and IKZF3, including their interaction term and the interaction between IKZF3 and vitamin D. Since IKZF3 forms homodimers and heterodimers with IKZF1, we included IKZF1 in the model. We adjusted the model for maternal asthma status and child race.

## Mice

All animal studies were approved by the institutional animal care committee. Mice were maintained in a virus- and parasite-free animal facility under a 12-hr light, 12-hr dark cycle. C57BL/6J WT mice were bred and housed at the MGH animal facility. Vitamin D-deficient mice (C57BL/6J) were bred and maintained in a UV-free environment and weaned onto a similar diet lacking vitamin D metabolites (TD97340; Harlan Teklad) that results in undetectable circulating 25-hydroxyvitamin D levels (*Sakai et al., 2001*). *Vdr*⁻ᐟ⁻ mice (Vdᵗᵐ¹ᴹᵇᵈ/ᴶ), on C57BL/6J background, were bred and housed at the MGH animal facility, weaned at 18 days on a diet that maintains normal mineral ion homeostasis in the absence of VDR signaling (2% calcium, 1.25% phosphorus, 20% lactose supplemented diet, TD96348;

Harlan Teklad, Madison WI) (*Li et al., 1998*). BALB/c male and female mice were purchased from Charles River Laboratories at 6–8 weeks of age and fed with diets supplemented with varying amounts of vitamin D3 (TD97340 was used as base diet and supplemented with either 400 IU vitamin D/kg, 1000 IU vitamin D/kg, or 4000 IU vitamin D/kg). Mice were bred on the specific diets for two generations to assure stabilization of vitamin D levels. Resulting F2 litters were used for the experiments. Mice were housed under standard conditions with free access to rodent chow and water. All animal experiments were carried out in accordance with NIH Guidelines for the Care and Use of Laboratory Animals as well as guidelines prescribed by the Institutional Animal Care and Use Committee (IACUC) at Brigham and Women's Hospital (2016N000357), Massachusetts General Hospital (2004N000113), and Harvard Medical Area (05115) (AAALAC 1729).

## HDM-induced airway inflammation

Mice were exposed to *D. pteronyssinus* extracts (Greer Laboratories, 10.52 EU/mg endotoxin) via the intranasal application of HDM (25 µg in saline) on days 1–3 and 8–15. Control mice received equal volumes of saline (Hospira Inc, Lake Forest, IL, USA). Twenty-four hours after the final challenge, mice were sacrificed, and specimens sampled.

## Bronchoalveolar Lavage

Bronchoalveolar Lavage (BAL) was performed with 2 × 1 ml of phosphate-buffered saline (PBS)/0.02% bovine serum albumin. The total number of leukocytes was determined using a TC20 Automated Cell Counter (Bio-Rad Systems). To differentiate cell types, cytospins were prepared and stained with Kwik-Diff solutions as per the manufacturer's instructions (Fisher Scientific) and 200 immune cells per slide were counted.

## Histological staining

Lungs were perfusion fixed with zinc fixative (BD Pharmingen, San Diego, CA) Tissues were paraffin embedded, divided into sections, and stained with hematoxylin and eosin by MGH DF/HCC Specialized Histopathology Services Core Histology Core. After paraffin embedment, 3 µm sections were stained with hematoxylin and eosin.

## Measurements of serum immunoglobulins

Blood samples were collected 24 hr after the final aerosol challenge and serum was prepared. Total IgE levels were measured by ELISA (Chondrex) according to the manufacturer's instructions.

## Dissociation of mouse tissues, in vitro polarization of CD4 Th subsets and flow cytometry analyses

Lungs were perfused via the right heart ventricle with PBS, excised, and digested with collagenase D (2 mg/ml; Roche) in RPM 1640 (Sigma-Aldrich) for 30 min at 37°C. The cell suspension was washed twice with PBS. Erythrocytes were removed by hypotonic lysis. Cells were washed twice with PBS and sieved with 30 µm cell strainers (Miltenyi Biotec). The resulting single-cell suspension was used to detect leukocyte populations (CD45 BV421 [clone: 104; Biolegend], CD4 PerCPeFluor710 [RM4-5; eBioscience], CD8a APC-Cy7 [53-6.7; Biolegend], CD19 PE-Cy7 [6D5; Biolegend]) and CD4+Foxp3+ Treg (CD45 BV421, CD4 PerCPeFluor710, Foxp3 PE [FJK-16s; Thermo Fisher]), VDR AlexaFluor647 (D2K6W; Cell Signaling Technologies), Aiolos PE (8B2; Biolegend) and cytokine production by Th cells (see below). Spleens were excised and mononuclear cells were isolated using a syringe plunger and 100 µm cell strainers. Red blood cells were removed by hypotonic lysis. Tonicity was restored by the addition of RPMI 1640 (Sigma-Aldrich). The resulting cell suspension was washed twice with PBS and used for the isolation of naive CD4+ T cells according to the manufacturer's instructions (Naive T Cell Isolation Kit II; Miltenyi Biotec). Naive CD4+ T cell purity was assessed by staining for CD4 PerCPeFluor710 (clone: RM4-5), CD44 FITC (IM7), and CD62L PE (mel-14). Naive T cells were activated with plate bound anti-CD3ε (5 µg/ml, clone: 145–2 C11, BD Biosciences) and soluble anti-CD28 (0.5 µg/ml, clone: 37.51, BD Biosciences). Th-subset polarization was induced by exposing naive CD4+ T cells to following culture conditions for 3 days. Th1 cells: recombinant murine IL-12 (rmIL-12) (10 ng/ml; Peprotech), anti-IL-4 (5 µg/ml; BD Biosciences); Th2 cells: rmIL-4 (20 ng/ml; Peprotech), anti-IFN-γ (5 µg/ml; BD Biosciences), and anti-IL-12 (5 µg/ml; BD Biosciences); Treg: rmTGFβ1 (2 ng/

ml; Peprotech), anti-IL-4 (5 µg/ml), anti-IFN-γ (5 µg/ml) for 3 days. Proliferation of Th1 and Th2 cells was induced using rmIL-2 (20 ng/ml; Peprotech) in the presence of Th-polarizing reagents for an additional 2 days. Polarization efficiency was verified by staining for intracellular cytokines (please see below) or Foxp3 expression (see above).

For intracellular cytokine staining, cells were stimulated with phorbol 12-myristate 13-acetate/ionomycin and brefeldin A (Cell Stimulation Cocktail; Tonbo Biosiences) for 4 hr. Cells were fixed with Fixation Buffer (Biolegend). Cells were permeabilized using 0.3% saponin in Fluorescence-activated cell sorting (FACS) buffer (0.5% FBS in PBS). Antibody staining was performed for 30 min in the dark using CD4, IFN-γ (XMG1.2-FITC), IL-10 (JES5-16E3-PE), IL-13 (eBio13A-eFluor660), and IL-17A (TC11-18H10.1-PE-Cy7). See *Supplementary file 1 and 2* for a full list of antibodies and reagents. Cells were washed twice with permeabilization buffer and suspended in FACS buffer. For analysis, 30,000 counts (in the CD4+ gate) were recorded on a FACSFortessa (BD Biosciences) and analyzed with FlowJo software (v10; Tree Star).

## Western blot analyses

Cells were lysed in 50 mmol/l Tris/HCl (pH 7.5), 150 mmol/l NaCl, 1% NP-40, and 1× protease and phosphatase inhibitor mix (both Roche) for 15 min on ice. Lysates were clarified, and total protein concentrations were measured with BCA (Pierce). Proteins (20 µg) were separated on gradient gels (Mini-PROTEAN TGX; Bio-Rad) and transferred on a Immun-Blot PVDF membrane (Bio-Rad). Unspecific binding was blocked using 5% milk powder in TBST (Tris-buffered saline) for 30 min. Anti-VDR (D2K6W, 1:1000; Cell Signaling Technologies, CST) was incubated in 0.5% milk powder in TBST overnight at 4°C. Membranes were washed three times with TBST and incubated with secondary anti-rabbit IgG, horseradish peroxidase (HRP)-linked Antibody (#7074; CST) for 2 hr. To control for equal loading, membranes was probed for β-actin (AC-15; 1:10,000; Sigma-Aldrich) expression for 2 hr at room temeperature (RT). Protein bands were visualized with Luminol reagent (Pierce) using the ChemiDoc XRS+ system (Bio-Rad).

## Immunofluorescence staining for confocal microscopy

In vitro polarized Th2 cells (see above) were stimulated with 100 nm calcitriol (in ethanol) or ethanol [equal vol/vol] for the final 24 hr of culture. Cells were washed twice with PBS and spun onto poly-l-lysine-coated glass slides. After drying, cells were fixed with 4% paraformaldehyde in PBS for 15 min. Cells were permeabilized using 0.2% Triton X-100 in PBS for 20 min. Cells were washed three times for 5 min with PBS and incubated with Image-iT FX Signal Enhancer (Thermo Fisher Scientific) for 30 min at RT. After three consecutive washes with PBS, unspecific binding was blocked using blocking buffer (PBS/5% goat serum/0.3% Triton X-100) for 60 min at RT. Primary antibodies directed against CD4-FITC (clone; 1:100, BD Biosciences), VDR (D2K6W; 1: 200, CST) were incubated in a humidified chamber over night at 4°C. After three washes with PBS, slides were incubated with the secondary reagent, anti-rabbit-Cy3 (1:500; Thermo) for 30 min at RT in the dark. After three more washes with PBS, slides were mounted with ProLong Gold with DAPI according to the manufacturer's instructions (Invitrogen). Images were acquired in a single z-plane using a Zeiss immunofluorescence microscope with a ×40 objective.

## Total RNA isolation, quality control, and RNA-sequencing analysis

Total RNA was isolated using TRIzol Reagent (Invitrogen) according to the manufacturer's instructions. Quality was assessed using the Nanodrop 8000 spectrophotometer. Sequencing libraries were constructed with the TruSeq Stranded Total RNA Library Prep Globin Kit (Illumina). Sequencing was performed using a HiSeq. 2500 instrument (Illumina). Trimmed reads were mapped to the GRCm38 reference genome using STAR (*Dobin et al., 2013*). Read counts were computed with htseq (*Anders et al., 2015*). Data were normalized and differential expression was analyzed with the R package DESeq2 (v.1.34.0) (*Love et al., 2014*) using a false discovery rate (FDR) <0.05.

## Pathway enrichment, overrepresentation, and GSEA

KEGG pathway, overrepresentation, and gene set enrichment analyses were performed with the 'enrichKEGG', 'enrichr', and 'GSEA' function in the R package 'clusterProfiler' (v.3.10.1) with default

settings (*Yu et al., 2012*). p values were adjusted using the Benjamini–Hochberg (BH)-FDR correction and enriched terms with a p-adj. <0.05 selected.

## Network analysis in the mice

RNA-sequencing data extracted from murine Th2 cells were analyzed using the R package DESeq2 (v.1.22.2) (*Love et al., 2014*). Raw counts were normalized and transformed for variance stabilization using the DESeq2 function rlog (regularized-logarithm transformation). Batch effects were removed using the function removeBatchEffect of the R package LIMMA (*Ritchie et al., 2015*). Gene with counts lower than 5 in at least one sample were removed for downstream analyses. Differential expression analysis was performed between control Th2 cells versus calcitriol-stimulated Th2 cells using DeSeq2. The design matrix was built to consider batch effects derived from the RNA isolation process. DESeq2-independent filtering was applied (*Love et al., 2014*) and p values were corrected for multiple hypothesis testing by using the BH-FDR adjustment (*Benjamini et al., 2001*).

The PPI network used in our network analysis was developed by *Silverbush and Sharan, 2019*. As described in the original paper, this human PPI network was reconstructed by integrating large-scale PPI databases, drug response repositories, and cancer genomics data. Only the largest connected component of the network was considered, resulting in an interactome of 15,500 proteins and 234,585 consensus-oriented interactions. Conversion between murine and human gene IDs was performed using the BioMart conversion tool (*Smedley et al., 2015*). We also assumed a one-to-one correspondence between genes and their protein products in the entire analysis.

To identify the active signaling paths in control and calcitriol-stimulated Th2 cells, we modeled the transmission of the molecular signal as a sequential path on the PPI network (called a network path) that starts from a receptor, connects the receptor to multiple intermediate proteins, and leads to a transcription factor. Receptors and transcription factors were selected from a predefined list based on their role in Th2 cell biology and IL-2 expression (*Figure 4—source data 1*). CD3, CD28, IL-4R, and IL-13R were selected based on their role in Th2 differentiation. The IL-2R complex was included due to being a target of vitamin D signaling. TGFβRs were selected based on the immunomodulatory activity of this signaling path in CD4+ T cells. role in mediating immunoregulation. Using this subset of receptors and transcription factors as input for the implementation of our network analysis, we computed all the shortest paths connecting each receptor–transcription factor pair on the PPI network. For both control and calcitriol-stimulated Th2 cells, we ranked each network path based on the average expression of the path's nodes. The top selected paths represent the most active downstream processes in Th2 cells in each condition.

## mRNA isolation, reverse transcription, and real-time RT-PCR

Total RNA from in vitro polarized Th2 cells was isolated with the RNeasy Mini Kit (QIAGEN). First-strand cDNA synthesis was performed with the Omniscript kit (QIAGEN). Quantitative real-time PCR was performed with the SsoFast EvaGreen Supermix Kit (Bio-Rad) on a AriaMx Real-time PCR System (Agilent) as per the manufacturer's instructions. Primers sequences used in this study are (*Supplementary file 3*): *Gsdma* (sense: 5′-TGC TGT CAA AGG ACG CTA TG-3′; antisense: 5′-TCA ACC AGC TTC AGT TGC AC-3′), *IL-2* (sense: 5′-CCC ACT TCA AGC TCC ACT TC-3′; antisense: 5′-ATC CTG GGG AGT TTC AGG TT-3′), *Ikfz3* (5′-TGT CAC CTC TGC AAC TAC GC-3′; 5′-TTC CGC AGA ACT CAC ACT TG-3′), *Ormdl3* (sense: 5′-CAC ACG GGT GAT GAA C AG TC-3′; antisense: 5′-CTT TGC CTT GGT CTG GAG TC-3′), *Stat3* (sense: 5′-GAG GAG CTG CAG AAA GT-3′; antisense: 5′-TCG TGG TAA ACT GGA CAC CA-3′), *Stat5a* (sense: 5′-TAC ATG GAC CAG GCT CCT TC-3′; antisense: 5′-GTC AAA CTC GCC ATC TTG GT-3′), and *Stat5b* (sense: 5′-TGT GGA TAC AGG CTC AGC AG-3′; antisense: 5′-TGG GTG GCC TTA ATG TTC TC-3′). Relative gene expression levels were calculated with the $2^{-\Delta\Delta Ct}$ method, with normalization to the murine ribosomal L32 as the housekeeping gene. Data are presented as fold-induction over the respective control groups (vehicle or WT). All standard procedures were performed according to the manufacturer's instructions.

## Quantification and statistical analysis

For each figure the number of replicates per experiment is indicated in the corresponding figure legend. In the figures, mean and standard error of the mean (SEM) are presented and error bars represent the mean ± SEM. Two group comparisons were analyzed using unpaired two-tailed Student's

*t*-tests (parametric data) and Mann–Whitney *U*-test (non-parametric data). Multiple group comparisons were analyzed with Kruskal–Wallis and one-way analysis of variance (ANOVA) test with Tukey's post hoc analysis. In vivo experiments were analyzed using mixed-effect analysis or two-way ANOVA test with Holm–Šidák's post hoc analysis (factors: genotype and exposure, diet and exposure). Prism 9.2 (GraphPad Software) was used to calculate statistics. p values of less than 0.05 were considered statistically significant.

## Acknowledgements

We wish to thank the VDAART trial participants and clinical staff for their critical work on the VDAART study. We wish to thank also our Channing colleagues for their comments.

## Additional information

### Competing interests

Scott T Weiss: receives royalties from UpToDate and is an investor in Histolix. The other authors declare that no competing interests exist.

### Funding

| Funder | Grant reference number | Author |
|---|---|---|
| National Heart, Lung, and Blood Institute | 5P01HL132825 | Scott T Weiss |
| National Heart, Lung, and Blood Institute | 5UH3OD023268 | Augusto A Litonjua |
| National Heart, Lung, and Blood Institute | 5K25HL150336 | Arda Halu |
| National Heart, Lung, and Blood Institute | 1K25HL168157 | Margherita De Marzio |
| National Heart, Lung, and Blood Institute | 1K01HL166705 | Enrico Maiorino |
| German Research Foundation | KI 1868/3-1 | Ayse Kilic |
| National Heart, Lung, and Blood Institute | 1R01HL171141 | Arda Halu |

The funders had no role in study design, data collection and interpretation, or the decision to submit the work for publication.

### Author contributions

Ayse Kilic, Conceptualization, Resources, Data curation, Formal analysis, Supervision, Funding acquisition, Validation, Investigation, Visualization, Methodology, Writing – original draft; Arda Halu, Data curation, Formal analysis, Funding acquisition, Visualization, Methodology, Writing – original draft, Co-first author/equal contribution; Margherita De Marzio, Enrico Maiorino, Formal analysis, Funding acquisition, Visualization, Methodology, Writing - review and editing; Melody G Duvall, Thayse Regina Bruggemann, Joselyn J Rojas Quintero, Hooman Mirzakhani, Ayse Özge Sungur, Janine Koepke, Taiji Nakano, Hong Yong Peh, Nandini Krishnamoorthy, Raja-Elie Abdulnour, Formal analysis, Investigation; Robert Chase, Data curation, Investigation; Katia Georgopoulos, Marie Demay, Investigation, Methodology; Augusto A Litonjua, Formal analysis, Funding acquisition, Investigation, Writing - review and editing; Harald Renz, Bruce D Levy, Supervision, Investigation, Writing - review and editing; Scott T Weiss, Conceptualization, Supervision, Funding acquisition, Investigation, Methodology, Writing – original draft

### Author ORCIDs

Ayse Kilic http://orcid.org/0000-0001-5362-2127
Arda Halu https://orcid.org/0000-0001-6217-790X

Margherita De Marzio ⓘ http://orcid.org/0000-0003-1821-208X
Melody G Duvall ⓘ https://orcid.org/0000-0002-4173-3628
Ayse Özge Sungur ⓘ https://orcid.org/0000-0002-9272-9717
Hong Yong Peh ⓘ http://orcid.org/0000-0001-6098-9381

### Ethics

Clinical trial registration NCT00920621.

The design, eligibility, and initial results of the trial have been published (11–17). Pregnant women were recruited from three clinical sites across the United States – Boston Medical Center, Boston, MA; Washington University at St. Louis, St. Louis, MO; and Kaiser Permanente Southern California Region, San Diego, CA. The Data Coordinating Center was based in the Channing Division of Network Medicine, Brigham and Women's Hospital, Boston, MA. Eligible participants were women between the ages of 18 and 39 years, who presented between the estimated gestational ages of 10 and 18 weeks; who had a history of asthma, eczema, or allergic rhinitis, or whose partner (biologic father of the child) had a history of asthma, eczema, or allergic rhinitis; who was a non-smoker; and who was English- or Spanish-speaking, with intent to participate for 4 years (up to the third birthday of the child). The VDAART study protocol was approved by the Institutional Review Boards at each participating institution and at the Brigham and Women's Hospital. All women provided a written informed consent. The trial was registered with clinical trials.gov: clinicaltrials.gov Identifier: NCT00920621.

All animal experiments were carried out in accordance with NIH Guidelines for the Care and Use of Laboratory Animals as well as guidelines prescribed by the Institutional Animal Care and Use Committee (IACUC) at Brigham and Women's Hospital (2016N000357), Massachusetts General Hospital (2004N000113), and Harvard Medical Area (05115) (AAALAC 1729).

Reviewer #1 (Public Review): https://doi.org/10.7554/eLife.89270.4.sa1
Reviewer #2 (Public Review): https://doi.org/10.7554/eLife.89270.4.sa2
Author response https://doi.org/10.7554/eLife.89270.4.sa3

## Additional files

### Supplementary files

• Supplementary file 1. List of antibodies used.

• Supplementary file 2. List of reagents used.

• Supplementary file 3. List of primer sequences used.

• MDAR checklist

### Data availability

All data generated or analyzed during this study are included in the manuscript and supporting files.

The following previously published datasets were used:

| Author(s) | Year | Dataset title | Dataset URL | Database and Identifier |
|---|---|---|---|---|
| Neme A, Seuter S, Carlberg C | 2017 | VDR ChIP-seq in human monocytic THP-1 cells | https://www.ncbi.nlm.nih.gov/geo/query/acc.cgi?acc=GSE89431 | NCBI Gene Expression Omnibus, GSE89431 |
| Ramagopalan SV | 2010 | Genome-wide analysis of vitamin D receptor binding by ChIP-Seq | https://www.ncbi.nlm.nih.gov/geo/query/acc.cgi?acc=GSE22484 | NCBI Gene Expression Omnibus, GSE22484 |

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
