## [Editor Report · eLife assessment]

The effect of Vitamin D supplementation in reducing asthma via anti-inflammatory mechanisms is a topic of wide interest, with somewhat conflicting published data. Here, bioinformatic approaches help to identify a role of VDR in inducing the expression of the key regulator Ikzf3, which possibly suppresses the IL-2/STAT5 axis, consequently blunting the Th2 response and mitigating allergic airway inflammation. These are **important** findings based on **convincing** evidence.

---

## [Referee Report · Reviewer #1 (Public Review)]

The association of vitamin D supplementation in reducing Asthma risk is well studied, although the mechanistic basis for this remains unanswered. In the presented study, Kilic and co-authors aim to dissect the pathway of Vitamin D-mediated amelioration of allergic airway inflammation. They use initial leads from bioinformatic approaches, which they then associate with results from a clinical trial (VDAART) and then validate them using experimental approaches in murine models. The authors identify a role of VDR in inducing the expression of the key regulator Ikzf3, which possibly suppresses the IL-2/STAT5 axis, consequently blunting the Th2 response and mitigating allergic airway inflammation.

The major strength of the paper lies in its interdisciplinary approach, right from hypothesis generation, and linkage with clinical data, as well as in the use of extensive ex vivo experiments and in vivo approaches using knock-out mice. The study presents some interesting findings including an inducible baseline absence/minimal expression of VDR in lymphocytes, which could have physiological implications and needs to be explored in future studies.

The study presents a potential for further dissection of relevant pathophysiological pathways to explain certain seemingly associative results, and allow for a more effective translation.

Several results in the study suggest multiple factors and pathways influencing the phenotype seen, which could be explored in the future. The inferences of this study also need to be read in the context of the different sub-phenotypes and endotypes of Asthma, where the Th2 response may not be predominant. While this does not undermine the importance of this elegant study, it is essential to emphasise a holistic picture while interpreting the results.

---

## [Referee Report · Reviewer #2 (Public Review)]

Summary:

This study seeks to advance our knowledge of how vitamin D may be protective in allergic airway disease using both adult and neonatal mouse models. The rationale and starting point are important human clinical, genetic/bioinformatic data, with a proposed role for vitamin D regulation of 2 human chromosomal loci (Chr17q12-21.1 and Chr17q21.2) linked to risk of immune-mediated/inflammatory disease. The authors have historically made significant contributions to this work specifically in airway disease/asthma. They now link these data to propose a role for vitamin D in regulating IL-2 in Th2 cells implicating genes associated with these loci in this process.

Strengths:

Here the authors draw together evidence from multiple interdisciplinary lines of investigation to propose that amongst murine CD4+ T cell populations, Th2 cells express high levels of VDR, and that vitamin D regulates many of the genes on the chromosomal loci identified to be of interest, in these cells. The bottom line is the proposal that vitamin D, via Ikfz3/Aiolos, suppresses IL-2 signalling in Th2 cells. This is a novel concept and whilst the availability of IL-2 and the control of IL-2 signalling is generally thought to play a role in the capacity of vitamin D to modulate both effector and especially regulatory T cell populations, this study provides new insights.

Weaknesses:

Ultimately the data are associative, nevertheless this study makes an important and innovative contribution to our understanding of the mechanism whereby vitamin D may beneficially control immune/inflammatory disease, specifically Th2 driven allergic airway inflammation. Future work advancing these studies, including in humans, are awaited with interest.

Wider impact: Maternal 17q21 genotype has an important influence on the protective effects of high dose vitamin D3 supplementation in pregnancy against the development of asthma/recurrent wheeze in her offspring. The current study provides exciting mechanistic data that may underpin this important observation.

---

## [Author Response]

The following is the authors’ response to the previous reviews.

**Reviewer #1 (Public Review):**
The association of vitamin D supplementation in reducing Asthma risk is well studied, although the mechanistic basis for this remains unanswered. In the presented study, Kilic and co-authors aim to dissect the pathway of Vitamin D-mediated amelioration of allergic airway inflammation. They use initial leads from bioinformatic approaches, which they then associate with results from a clinical trial (VDAART) and then validate them using experimental approaches in murine models. The authors identify a role of VDR in inducing the expression of the key regulator Ikzf3, which possibly suppresses the IL-2/STAT5 axis, consequently blunting the Th2 response and mitigating allergic airway inflammation.The major strength of the paper lies in its interdisciplinary approach, right from hypothesis generation, and linkage with clinical data, as well as in the use of extensive ex vivo experiments and in vivo approaches using knock-out mice. The study presents some interesting findings including an inducible baseline absence/minimal expression of VDR in lymphocytes, which could have physiological implications and needs to be explored in future studies.However, the study presents a potential for further dissection of relevant pathophysiological parameters using additional techniques, to explain certain seemingly associative results, and allow for a more effective translation.Several results in the study suggest multiple factors and pathways influencing the phenotype seen, which remain unexplored. The inferences of this study also need to be read in the context of the different sub-phenotypes and endotypes of Asthma, where the Th2 response may not be predominant. While this does not undermine the importance of this elegant study, it is essential to emphasize a holistic picture while interpreting the results.
**Reviewer #2 (Public Review):**
Summary:This study seeks to advance our knowledge of how vitamin D may be protective in allergic airway disease in both adult and neonatal mouse models. The rationale and starting point are important human clinical, genetic/bioinformatic data, with a proposed role for vitamin D regulation of 2 human chromosomal loci (Chr17q12-21.1 and Chr17q21.2) linked to the risk of immune-mediated/inflammatory disease. The authors have made significant contributions to this work specifically in airway disease/asthma. They link these data to propose a role for vitamin D in regulating IL-2 in Th2 cells implicating genes associated with these loci in this process.Strengths:Here the authors draw together evidence form. multiple lines of investigation to propose that amongst murine CD4+ T cell populations, Th2 cells express high levels of VDR, and that vitamin D regulates many of the genes on the chromosomal loci identified to be of interest, in these cells. The bottom line is the proposal that vitamin D, via Ikfz3/Aiolos, suppresses IL-2 signalling and reduces IL-2 signalling in Th2 cells. This is a novel concept and whilst the availability of IL-2 and the control of IL-2 signalling is generally thought to play a role in the capacity of vitamin D to modulate both effector and especially regulatory T cell populations, this study provides new data.Weaknesses:Overall, this is a highly complicated paper with numerous strands of investigation, methodologies etc. It is not "easy" reading to follow the logic between each series of experiments and also frequently fine detail of many of the experimental systems used (too numerous to list), which will likely frustrate immunologists interested in this. There is already extensive scientific literature on many aspects of the work presented, much of which is not acknowledged and largely ignored. For example, reports on the effects of vitamin D on Th2 cells are highly contradictory, especially in vitro, even though most studies agree that in vivo effects are largely protective. Similarly other reports on adult and neonatal models of vitamin D and modulation of allergic airway disease are not referenced. In summary, the data presentation is unwieldy, with numerous supplementary additions, that makes the data difficult to evaluate and the central message lost. Whilst there are novel data of interest to the vitamin D and wider community, this manuscript would benefit from editing to make it much more readily accessible to the reader.Wider impact: Strategies to target the IL-2 pathway have long been considered and there is a wealth of knowledge here in autoimmune disease, transplantation, GvHD etc - with some great messages pertinent to the current study. This includes the use of IL-2, including low dose IL-2 to boost Treg but not effector T cell populations, to engineered molecules to target IL-2/IL-2R.
**Recommendations for the authors:**

**Reviewer #1 (Recommendations For The Authors):**
In the revised manuscript, the authors have addressed a significant number of concerns raised. The restructuring and incorporation of a number of discussion points have improved the readability. Moreover, the authors have also incorporated some more figures to address certain questions raised.However, the authors could reconsider a few more points which would improve the readability of the manuscript.For e.g.1. While it is appreciated that the authors have provided the schematic of the study design for the VDAART trial, the visualization for the RNA-seq analysis may be helpful.

We have created a visualization of the workflow for the RNA seq analysis as part of Figure 1 – figure supplement 1C.

2. Quantification of images would not require any additional experiments, yet can reinforce the results with objectivity.

We appreciate this comment. We chose to display histology images to allow a glimpse at the inflammatory condition in the lung tissue. For histological quantification, lung tissue should have been harvested and analyzed in a systematic and randomized way as well as in sufficient animal numbers to allow statistical analyses. This has not been done for these mouse models since the focus was in analyzing cytokine production by lung tissue CD4+ T cells as the driver of inflammation.

3. The authors have not addressed the discrepancy of the sample sizes in the experiments. Some dot plots still don't match the legends, and there is a wide variation in the numbers chosen for different experiments and different groups in the same experiments.

We appreciate the thorough screening of our manuscript and apologize for this oversight. We corrected the errors in the respective figure legends.

The in vivo experiments comprise studies performed in (A) VDR-KO mice and (B) WT mice fed with vit-D supplemented chow.

Sample size calculations for the mouse models of allergic airway inflammation based on BAL cell numbers revealed a minimum of n=8 per group for correct statistical analysis. In both experimental settings, the respective mouse lines were bred in the mouse facilities of MGH (A) and BWH (B). Depending on the litter sizes, additional mice were added in the HDM group, since bigger variability was expected in this group than the saline group.

Intracellular CD4+ cytokine staining was performed for all mice, however some stainings failed and could not be reliably interpreted and were therefore excluded.

**Reviewer #2 (Recommendations For The Authors):**
The authors have largely replied to the reviewer comments, amended some noted typos & figure legend issues, as well as discussed the reviewers concerns in text and in their rebuttal.The data presented are novel and of significant interest, conceptually moving this field forward, but in this reviewer's opinion reflect one pathway, of likely several, linked to protective effects of vitamin D on airway disease. This reviewer recommends a further slight editing of the text to present this broader scenario.i) Treg cells are highly dependent on IL-2 (both Foxp3+ and IL-10+ cells, not always the same population), constitutively express the IL-2R, and there is already a significant literature regarding vitamin D and IL-10/Treg in control of immune-mediated conditions. A simple statement acknowledging this and that there are likely more than one mechanisms by which vitamin D may regulate allergic airway disease (directly or indirectly) would be appreciated - this is no way detracts from the novelty and contribution of the current findings.

We thank the reviewer for this suggestion. We have added the following statement to the manuscript (lines 623-625):

“Additional pathways, including the induction of IL-10 production by CD4+ T cells as well as a direct induction of Foxp3+ T reg cells could have further contributed to the observed protective effect of vitamin D supplementation (PMID: 21047796; 22529297).”

ii) More comprehensive referencing of earlier papers proposing effects of vitamin D in controlling Treg/IL-10 and dampening Th2 responses in mouse (and human) models(e.g. Taher, Y. A., van Esch, B. C. A. M., Hofman, G. A., Henricks, P. A. J. & van Oosterhout, A. J. M. 1alpha,25-dihydroxyvitamin D3 potentiates the beneficial effects of allergen immunotherapy in a mouse model of allergic asthma: role for IL-10 and TGF-beta. J. Immunol. 180, 5211-21 (2008). Vassiliou JE et al, 2014. Vitamin D deficiency induces Th2 skewing and eosinophilia in neonatal allergic airways disease. Allergy DOI10.1111/all.12465).

We have included the reference in the discussion section of our manuscript in lines 617-619:

“Similar findings regarding the effects of vitamin D in controlling Treg/IL-10 and dampening Th2 responses have been reported, e.g., in (PMID: 18390702) and in offspring of mice that had been subjected to vitamin D deficiency in the third trimester of their pregnancy (PMID: 24943330).”